# Evaluating Country-Scale Irrigation Demand Through Parsimonious Agro-Hydrological Modeling

Nike Chiesa Turiano *, Marta Tuninetti, Francesco Laio and Luca Ridolfi

Department of Environment, Land and Infrastructure Engineering (DIATI), Politecnico di Torino, Corso Duca degli Abruzzi, 24, 10129 Torino, Italy; marta.tuninetti@polito.it (M.T.); francesco.laio@polito.it (F.L.); luca.ridolfi@polito.it (L.R.)
*   Correspondence: nike.chiesa@polito.it

## Abstract

Climate change is expected to reduce water availability during cropping season, while growing populations and rising living standards will increase the global water demand. This creates an urgent need for national water management tools to optimize water allocation. In particular, agriculture requires targeted approaches to improve efficiency. Alongside field measurements and remote sensing, agro-hydrological models have emerged as a particularly valuable resource for assessing and managing agricultural water demand. This study introduces WaterCROPv2, a state-of-the-art agro-hydrological model designed to estimate national-scale irrigation water demand while effectively balancing accuracy with practical data requirements. WaterCROPv2 incorporates innovative features such as hourly time-step computations, advanced rainwater canopy interception modeling, detailed soil-dependent leakage dynamics, and localized daily evapotranspiration patterns based on meteorological data. Through comprehensive analyses, WaterCROPv2 demonstrates significantly enhanced reliability in estimating irrigation water needs across various climatic regions, particularly under contrasting dry and wet conditions. Validation against independent data from the Italian National Institute of Statistics (ISTAT) for maize cultivation in Italy in 2010 confirms the model's accuracy and underscores its potential for broader international applications. A spatial analysis further reveals that the estimation errors align closely with regional precipitation patterns: the model tends to slightly underestimate irrigation needs in the wetter northern regions, whereas it somewhat overestimates demand in the drier southern areas. WaterCROPv2 has also been used to analyze irrigation water requirements for maize cultivation in Italy from 2005 to 2015, highlighting its significant potential as a strategic decision-support tool. The model identifies optimal cultivation areas, such as the Pianura Padana, where the irrigation requirements do not exceed 200 mm for the entire maize growing period, and unsuitable regions, such as Salentino, where over 500 mm per season are required due to the local climatic conditions. In addition, estimates of the water volumes required for the current extent of maize cultivation show that the Pianura Padana region demands nearly three times the amount of water used in the Salentino area. The model has also been used to identify regions where adopting efficient irrigation technologies could lead to substantial water savings. With micro-irrigation currently covering less than 18% of irrigated land, simulations suggest that a complete transition to this system could reduce the national water demand by 21%. Savings could reach 30–40% in traditionally water-rich regions that rely on inefficient irrigation practices but are expected to be increasingly exposed to temperature increases and precipitation shifts. The analysis shows that those regions currently lacking adequate irrigation infrastructure stand to gain the most from targeted irrigation system investments but also highlights how incentives where micro-irrigation is already widespread can provide further 5–10% savings.

**Keywords:** agro-hydrological modeling; blue water; water management

## 1. Introduction

Water governance operates at multiple levels, from international organizations down to local consortia [1]. This multi-level approach is essential to guarantee that water policies and laws ensure both (water-dependent) food and energy security without undermining environmental protection [2,3]. In fact, nowadays, 40% of the world's food production comes from irrigated land, which, despite accounting for only 18% of the global cultivated land [4], accounts for from 60% [5] to 70% of the total freshwater withdrawals [6]. This pressure on freshwater bodies is expected to increase significantly, mainly due to the world's population growth [7], dietary changes towards more water-demanding food choices [8,9], and the need to buffer the increased vulnerability of cultivated lands to climate extremes [10]. Furthermore, future alterations in the hydrological cycle due to climate change [11–14] will likely impact the natural availability of freshwater resources, making effective water governance even more crucial [15]. This highlights the need for effective local water management even in water-rich areas where water exploitation has often been overlooked [16].

Under these conditions, there is an ongoing effort to improve national agricultural water management to reduce both agriculture's vulnerability to climatic variability and the stress on freshwater bodies and the associated ecosystem [17–19]. Effective national water management has the potential to regulate local water withdrawals from water bodies, preventing regional excessive use of irrigation water and reducing crop losses due to water stress [20,21].

To this aim, accurately evaluating irrigation demands, along with other water uses, is essential for correct water management and allocation [22]. Comprehensive national-scale evaluations of agricultural water use can provide pictures of the current water consumption across different regions. This aids in identifying areas where structural and management improvements are needed, thus supporting decision-making. Furthermore, reliable models can simulate various potential scenarios and predict the impact of different water management practices, providing valuable insights for investments and action plans.

In this context, agro-hydrological models are powerful decision-support tools for evaluating agricultural water requirements at various scales [16,23–26]. Numerous models have been developed over the years for this purpose (e.g., AquaCrop [27], SWAP [28], DSSAT [29], CropWat [30], APSIM [31], and CropSyst [32]), encompassing a wide range of modeling approaches, accuracy levels, and flexibility [33,34]. However, the use of most of these models is constrained by their data requirements and the spatial scales at which they can be applied. In fact, on one hand, highly complex models offer great accuracy and physical detail but are typically field-specific, computationally demanding, and require detailed input data that is often difficult to obtain [35]. It follows that these models are applied at the field scale but are hardly upscalable at the regional and national scales.

On the other hand, simpler large-scale models are adopted for simulations at the continental and global scales [36,37]. They require much less information than the previous class of models but may be too crude for agricultural water planning and management at the regional scale [38]. They often fail to account for variations in irrigation practices or soil diversity at the municipal scale, operate on a daily time scale, and approximate the modeling of the soil water balance.

In this framework, our aim is to propose a physically based agro-hydrological model that balances the complexity of detailed models with the lower data demand of large-scale models.

Our model describes all the main hydrological processes that determine agricultural water demand but, at the same time, requires data typically available at a municipal resolution rather than using a coarse spatial resolution as typically employed for regional and global models. By providing both quantitative outputs and spatially explicit maps, the model proves suitable for regional-scale water resource planning and management. Its accessibility and simplicity make the model usable by both specialists and non-experts, enabling process tracking, parameter adjustment, and critical evaluation of results.

The WaterCROPv2 model proposed herein builds upon WaterCROPv1 [39], introduced in 2015, enhancing it in several regards by incorporating new hydrological processes and refining the existing ones. Although the model presented in Tuninetti et al. was originally not named, it set the foundation for the present model. In agreement with its original developer, we refer to it here as WaterCROPv1.

To show the potential of the proposed model, we will (i) describe the physical processes considered and their modeling, (ii) show the typical WaterCROPv2 output, highlighting the advantages obtained with the updates in the crop water demand assessment, and (iii) focus on irrigated maize cultivation in Italy as an exemplifying case study. The case study allows us not only to demonstrate the reliability of WaterCROPv2 in assessing irrigation water demand but also to shed light on how the model can be used to analyze scenarios aimed at saving irrigation water and reducing the stress on freshwater resources.

## 2. Materials and Methods

### 2.1. Model

WaterCROPv2 is an agro-hydrological bucket model that simulates the crop and irrigation water demand during the growing season depending on soil, climate, crop, and irrigation system features. The bucket is designed as a single soil layer whose depth corresponds to the length of the plant roots $Zn$. As $Zn$ elongates during the plant growth, the control volume of the storage changes its depth accordingly. The bucket not only stores water but also receives and releases hourly volumes of water. The fluxes entering the storage are (see Figure 1) the effective precipitation ($P_{eff}$) and the crop blue water demand ($I_b$), while the exiting ones are the leakage ($L$) and the actual evapotranspiration ($ET_a$), also known as crop water demand. $P_{eff}$ is the precipitation $P$, which first decreased regarding the volume of drizzle water intercepted by the canopy, $T$, and secondly the runoff, $R$. All the variables used in the model are summarized in Table 1.

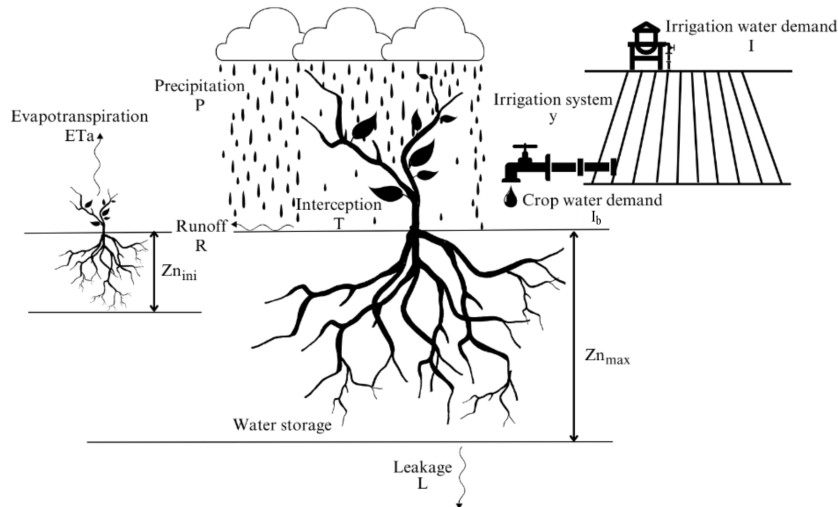

**Figure 1.** Water balance components of the bucket model: precipitation $P$, interception $T$, runoff $R$, evapotranspiration $ET_a$, crop water demand $I_b$, and leakage $L$. $Zn_{ini}, Zn_{max}, y$ stand for initial root depth, maximum root depth, and irrigation system, respectively.

**Table 1.** List of variables used in the paper.

| Variable | Description | Unit |
|---|---|---|
| d | Day | day number |
| Dr* | Critical depletion fraction | m |
| $ET_0$ | Potential evapotranspiration | mm/hour |
| $ET_a$ | Actual evapotranspiration | mm/hour |
| I | Irrigation | mm/day |
| $I_b$ | Blue water requirement | mm/day |
| $I_{mz}$ | Maize irrigation water demand | mm/day |
| h | Hour | hour |
| H | Time enlapsed since sunrise | hour |
| $k_c$ | Crop coefficient | - |
| $k_{c,fin}$ | Crop coefficient at the end of the growing season | - |
| $k_{c,ini}$ | Crop coefficeint at the beginning of the growing season | - |
| $k_{c,mid}$ | Crop coefficient at plant maturity | - |
| $k_s$ | Water stress coefficient | - |
| $k_{s,sat}$ | Saturated hydraulic conductivity | m/s |
| L | Leakage | mm/hour |
| lgp | Length of the growing period | days |
| N | Number of hours between sunrise and sunset | hour |
| q | Generic quantity | - |
| P | Precipitation | mm/hour |
| $P_{eff}$ | Effective Precipitation | mm/hour |
| R | Runoff | mm/hour |
| s | Relative soil moisture | - |
| $s_{fc}$ | Relative soil moisture at field capacity | - |
| sr | Sunrise time | hour of day |
| ss | Sunset time | hour of day |
| $SI_q$ | Sensitivity Index | - |
| t | time | hour |
| T | Canopy interception | mm/hour |
| $V_{soil}$ | Soil volume | $m^3$ |
| WC | Soil water content | m |
| $WC_{fc}$ | Soil water content at field capacity | m |
| $WC_{sat}$ | Soil water content at saturation | m |
| $WC_{th}$ | Soil water content at a chosen water content threshold | m |
| $WC_{wp}$ | Soil water content at wilting point | m |
| $WC_*$ | Soil water content at critical point | m |
| y | Irrigation system | - |
| $Zn_{ini}$ | Sowing depth | m |
| $Zn_{max}$ | Maximum Root depth | m |
| $\alpha$ | Irrigation inefficiency of the irrigation system | - |
| $\beta$ | Soil-dependent coefficient | - |
| $\theta_{fc}$ | Volumetric water content at field capacity | m/m |
| $\theta_{sat}$ | Volumetric water content at saturation | m/m |
| $\theta_*$ | Volumetric water content at critical point | m/m |

The mentioned input and output fluxes entail the temporal fluctuations in the water stored in the soil, namely the so-called soil water content, *WC*. The water balance equation for the soil layer reads

$$\frac{dWC(t)}{dt} = P(t) - T(P(t)) - R(WC(t)) - ET_a(WC(t)) - L(WC(t)) + I_b(WC(t)) \quad (1)$$

where the soil water content is expressed in mm, time *t* in hours, and fluxes in mm/hour. It is important to note that, as will be described further later on, $R, ET_a, T, L$, and $I_b$ are

functions of the soil water content. $WC$ can span from $WC = 0$ mm to the soil water content at saturation $WC_{sat}$, whose magnitude depends on the soil type and the root length $Zn$. Specific $WC$ values play a crucial role in regulating the input/output fluxes (Figure 2):

- $WC_{wp}$. It is the $WC$ at wilting point and sets the minimum water content for plants to survive;
- $WC_*$. It is the $WC$ at critical condition and is the threshold that controls whether evapotranspiration is at maximum and whether it is necessary to resort to irrigation;
- $WC_{fc}$. It is the $WC$ at field capacity, and, when $WC$ goes below $WC_{fc}$, the leakage process ends;
- $WC_{sat}$. It is the $WC$ at saturation, which controls the surface runoff formation.

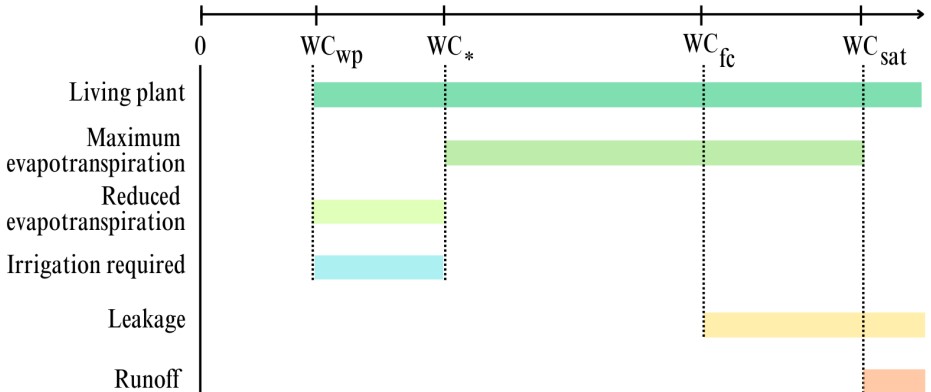

**Figure 2.** Ranges of possible water content (WC) values for different processes. Note that the light-blue bar of irrigation requirements shows the case of irrigation up to the critical level ($WC_*$).

The previously defined $WC$ thresholds are not fixed values but are proportional to the length of the plant roots, $Zn$, which elongate during the growing season: the further the roots grow into humid soil, the larger the amount of water (corresponding to the thresholds) contained in the control volume becomes.

The growing period is subdivided into four growing stages, namely initial phase (establishment, $lgp_1$), development stage (vegetative, $lgp_2$), mid-season (flowering, $lgp_3$), and late season (yield formation and ripening, $lgp_4$). Taking $Zn_{ini}$ as initial root length, $Zn_{max}$ as maximum length, and $d$ as the day along the growing period, $Zn$ is computed as [30]

$$Zn_t = \begin{cases} Zn_{ini} & \text{if } d = 1 \\ Zn_{ini} + \frac{Zn_{max} - Zn_{ini}}{lgp_1 + lgp_2} d & \text{if } d \in [lgp_1, lgp_2] \\ Zn_{max} & \text{if } d \in [lgp_3, lgp_4] \end{cases} \tag{2}$$

$Zn$ is assumed to always be in the vadose zone. It follows that there is no interaction between root zone and the water table. For a visualization of the elongation trend of $Zn$ during the growing period, refer to Figure A4 in Appendix B.

WaterCROPv2 is designed so that the continuous description of the water balance (Equation (1)) in time $t$ is discretized in hours that, every day, run from $h = 1$ to $h = 24$. This discretization implies that $ET_a$, $L$, and $P$, which naturally occur during or throughout the whole day, are discretized hourly. Every day, at the first hour ($h = 1$ of the day $d$), the bucket stores the same volume of water as the last hour of the previous day ($h = 24$ of day $d - 1$) plus $I_b$, computed as the amount of water needed to reach the chosen water content, i.e., $WC_d(h = 1) = WC_{d-1}(h = 24) + ET_{b,d-1}$.

**Evapotranspiration.** $ET_a$ takes into account plant transpiration and water evaporation from the soil. Each hour $h$ along the growing period, $ET_a$, is evaluated as

$$ET_a(h,d) = \begin{cases} ET_0(h,d)k_c(d)k_s(h) & \text{if } P(h) = 0 \\ 0 & \text{if } P(h) > 0 \end{cases} \tag{3}$$

where $ET_0$ is the reference evapotranspiration, $k_c$ is the crop coefficient, and $k_s$ is the water stress coefficient. As $ET_a$ rate is strongly driven by the gradient in relative humidity, $\Delta e$, between stomata and atmosphere, in case of rain events ($\Delta e \simeq 0$), $ET_a$ is assumed to shut down.

$ET_0$ is defined as the evapotranspiration of a hypothetical well-watered grass reference crop with fixed height, albedo, and surface resistance [30]. Databases usually provide daily $ET_0$ values in spite of transpiration being a diurnal process that shows a peak around mid-day. In order to account for daily plant physiology, in WaterCROPv2, $ET_0$ is modeled according to [40]

$$\begin{cases} ET_0(h,d) = \frac{ET_{0,d}(d)\pi\sin(\frac{\pi H}{N})}{2N} \\ N = ss - sr \\ H = h - sr \end{cases} \tag{4}$$

where $ET_{0,d}(d)$ is the daily evapotranspiration value corresponding to the day $d$, $N$ is the amount of hours spanning from sunrise $sr$ to sunset $ss$, and $H$ is the time elapsed since sunrise. Sunrise and sunset timing were defined, cell by cell, according to location (latitude and longitude) and day of the year.

The crop coefficient, $k_c$, is the coefficient that distinguishes the evapotranspiration rate of a specific crop from the one of the reference grass. $k_c$ varies during the growing season depending on the crop development in order to take into account changes in crop height and leaf areas. In particular, $k_c$ evolves in time as [30]

$$k_{c,t} = \begin{cases} k_{c,ini} & \text{if } d \in lgp_1 \\ \frac{k_{c,mid} - k_{c,ini}}{d - lgp_2}d & \text{if } d \in lgp_2 \\ k_{c,mid} & \text{if } d \in lgp_3 \\ \frac{k_{c,fin} - k_{c,mid}}{d - lgp_1 - lgp_2 - lgp_3}d & \text{if } d \in lgp_4 \end{cases} \tag{5}$$

where $k_{c,ini}$, $k_{c,mid}$, and $k_{c,fin}$ are $k_c$ at initial, mid, and final growing stages, respectively, and $d$ is the day along the growing period. Note that $k_c$ values and $lgp$s lengths depend on climate zone and crop type [30].

The water stress coefficient, $k_s \in [0,1]$, is the coefficient that specifies whether the plant is watered enough to evapotranspire at its potential. Thus, it is regulated by $WC$ as [30]

$$k_s(h) = \begin{cases} 0 & \text{if } WC \leq WC_{wp} \\ \frac{WC(h) - WC_{wp}}{WC_* - WC_{wp}} & \text{if } WC \in [WC_{wp}, WC_*] \\ 1 & \text{if } WC \geq WC_* \end{cases} \tag{6}$$

To have a visualization of $k_c$ trend during the growing period and of $k_s$ trend as function of $WC$, please, refer to Figures A4 and A5, respectively, in Appendix B.

If soil water content is above the critical value, $WC_*$, the plant is in stress-free conditions and evapotranspires at its maximum potential, $ET_a = ET_c$. Instead, whenever the water content is below the critical value, the plant is under stress and the evapotranspiration is lower than the potential one, $ET_a < ET_c$. This linear reduction is due to a gradual closure of the stomata that results in a progressive diminution in water uptake. If wilting point is reached, $WC = WC_{wp}$, water uptake stops and evapotranspiration ceases, $ET_a = 0$. Physical evaporation can take place below wilting point independently from plant transpiration. However, this phenomenon is not modeled here as this study focuses on irrigated crops that unlikely reach such conditions. Figure 3 reports an example of $ET_a$ evolution (purple line): the last hump shows the undisturbed daily trend, while the previous humps point out the effect of rain (green bins). Rain events cause the interruption of daily $ET_a$, but, if the rainwater has time to evaporate from the leaves, $ET_a$ activates again (second hump).

**Leakage.** Leakage is modeled with the following exponential law [41]:

$$L(s) = \begin{cases} \dfrac{K_s}{e^{\beta(1-s_{fc})}-1}\left[e^{\beta(s-s_{fc})}-1\right] & \text{if } s_{fc} < s \leq 1 \\ 0 & \text{if } s < s_{fc} \end{cases} \tag{7}$$

where $K_s$ is the saturated hydraulic conductivity, $\beta$ a soil-dependent coefficient, $s$ the relative soil moisture defined as $s = WC/WC_{sat}$, and $s_{fc}$ the relative soil moisture at field capacity. Coherently with the time discretization of water balance (Equation (1)), $L$ is evaluated with hourly steps.

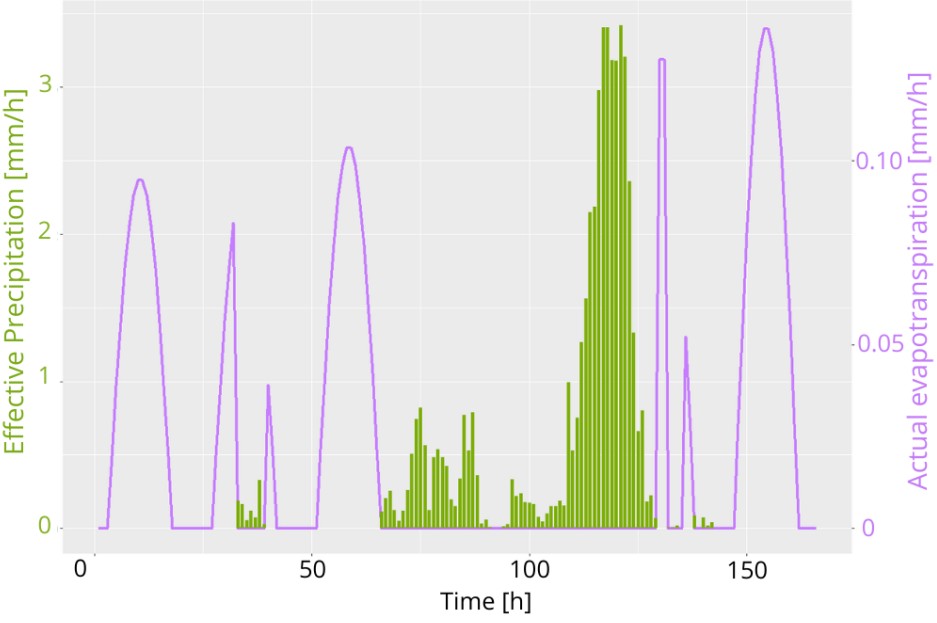

**Figure 3.** Two rain events are reported that stop the daily evapotranspiration process. The evapotranspiration behavior is shown for 6 days (from 28th to 35th day) in May of the initial stage of maize growth ($k_c \in [0.3–0.4125]$), during which the plant is not under stress ($k_s = 1$).

**Precipitation, interception, and runoff.** To avoid very small precipitation values that do not affect the soil hydrological balance but can instead introduce spurious noise in the system dynamics, a filter was introduced: a rainfall is considered a rain event only if it accumulates at least 0.01 mm/h. Canopy interception, $T(h)$, is included by subtracting, from each rain event, 0.5 mm for grasses and 2 mm for trees [41]. Another key point is when rainfalls can be considered as two separate rain events. In this work, a reasonable break between two subsequent events was set at 5 h.

Runoff, $R(h)$, is formed only if the effective precipitation, $P_{eff} = P(h) - T(h)$, brings in the control volume more water than the amount that can be stored. The overflowing water forms runoff as

$$\begin{cases} R(h) = P(h) - T(h) - WC_{av}(h) & \text{if } P_{eff} > WC_{av} \\ WC_{av}(h) = WC_{sat}(d) - WC(h) \end{cases} \tag{8}$$

**Irrigation.** WaterCROPv2 computes irrigation volumes $I$ in two steps: first, it computes the crop blue water demand, $I_b$, and secondly the irrigation volumes $I$ (called irrigation water demand). $I_b$ corresponds to the theoretical amount of water needed to keep the soil at a chosen water content threshold, $WC_{th}$, taking into account soil dynamics (leakage) and plant physiology (evapotranspiration). Generally, $WC_{th}$ is chosen so that the plant can evapotranspire at its potential: e.g., $WC_* < WC_{th} < WC_{FC}$. In WaterCROPv2, $I_b$ is applied, conventionally at $h = 24$ (but other choices are easily implementable) only if the soil water content is below the chosen water content, and only to reach such threshold. It follows that

$$I_b(d) = \frac{WC_{th}(d) - WC(h = 24)}{\Delta h} \tag{9}$$

Irrigation water demand $I$ is the actual amount of water that the farmer has to provide to the field. Such value depends on the irrigation systems as it encloses the irrigation system inefficiency. There are several ways of defining the irrigation system inefficiency $\alpha$ according to the application [5]. In this work, we neglect the inefficiency of the conveyance system that transports water from the water body to the field as we instead focus on the irrigation method present on the field (e.g., submersion, flow irrigation, sprinklers, or micro-irrigation). Thus, in this work, the irrigation inefficiency $\alpha$ ($\alpha > 1$) is defined, for each irrigation system, as the average ratio between the volumes of water provided to the field ($I$) and the one that has to infiltrate the rootzone ($I_b$), namely

$$I(d) = \alpha I_b(d) \tag{10}$$

**Flowchart.** To aid the reader in grasping the logic of the model, the flowchart of the operations implemented by the model is reported in Figure 4.

**Modifications of WaterCROPv2 with respect to version 1.** Note that the improvements in water demand modeling of WaterCROPv2 with respect to WaterCROPv1 include (i) the hourly timescale of the soil water balance, (ii) the use of hourly meteorological data instead of monthly precipitation, (iii) the inclusion of the canopy interception of rainwater, (iv) the simulation of leakage dynamics in relation to soil water content and soil characteristics, (v) a detailed description of the diurnal trend of evapotranspiration accounting also for location, day of the year, and the presence or absence of rain, and (vi) the computation of $I_b$ as the water that is required by the plant to be able to keep the field at a chosen water content. Finally, (vii) a key change is the introduction of the inefficiency factor of the irrigation system to take into account the additional water volume required to actually convey $I_b$ to the plants.

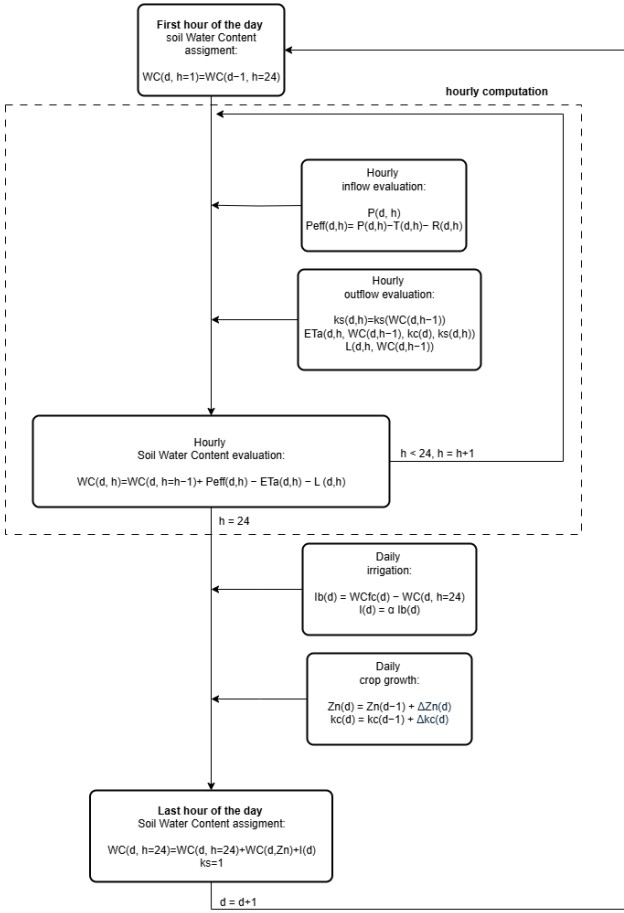

**Figure 4.** Flowchart of the model functioning.

## 2.2. Data

WaterCROPv2 works at hourly temporal resolution on grid cells whose size is determined by the input data resolution. All the required data must be homogenized at the same spatial and temporal resolution. The required input data fall into four categories: crop data, soil data, climate data, and irrigation data.

**Crop data.** Crop data characterize the crop in terms of growth and response to water stress. They are (i) daily reference evapotranspiration ($ET_0, d$), (ii) crop-specific growth coefficients ($kc_{ini}$, $kc_{mid}$, and $kc_{fin}$), (iii) length of the four growing period stages ($lgp_1$, $lgp_2$, $lgp_3$, and $lgp_4$), (iv) critical soil moisture ($WC_*$), (v) initial and maximum root depth ($Zn_{ini}$ and $Zn_{max}$), (vi) sowing and harvesting dates, and (vii) extension of irrigated areas. Notice that sowing and harvesting dates, length of the growing stages, and root depth depend on cultural conditions and on the local climate zone.

It is noteworthy that, in the literature, it is easier to retrieve values of critical depletion fraction, $Dr_*$, rather than $WC_*$. $Dr_*$ is defined as the critical soil water shortage with respect to field capacity and is related to $WC_*$ as follows:

$$Dr_* = WC_{fc} - WC_* \tag{11}$$

**Soil data.** The description of runoff and leakage processes requires: (i) saturated hydraulic conductivity ($K_s$), (ii) volumetric water content at field capacity ($\theta_{fc}$), (iii) volumetric water content at saturation ($\theta_{sat}$), and (iv) the fitting coefficient ($\beta$). Note that the volumetric water content $\theta$ is defined as the ratio of water volume $WC$ to soil volume $V_{soil}$ (e.g., $\theta_{fc} = WC_{fc}/V_{soil}$).

**Climate data.** Major players in plant phenology are local climate and precipitation. Thus, (i) climate zone, (ii) hourly precipitation time series, and (iii) geographical coordinates have to be provided for in the studied area.

**Irrigation data.** WaterCROPv2 determines the irrigation water demand, $I$, from the blue water demand, $I_b$ (Equation (12)). Thus, the inefficiency factor $\alpha$ of the used irrigation systems has to be provided.

## 3. Results

WaterCROPv2 was developed to evaluate in a reliable and functional way irrigation water demand at seasonal and regional scales. As an example, we report and analyze the outputs of the employment of WaterCROPv2 at the Italian national scale. In Section 3.1, firstly, the mean maize irrigation demand for the years 2005–2015 obtained by Water-CROPv2 is compared to the evaluation obtained from the previous version, WaterCROPv1, to show the effect of modeling improvements; secondly, the model reliability is assessed using independent data of irrigation water provided by the Italian National Institute of Statistics, ISTAT [42], for 2010. Lastly, Section 3.2 presents some possible applications of the WaterCROPv2 model as a decision-making tool using country-scale mean results for maize cultivation in Italy. Maize is, in fact, one of the most water-demanding crops, and, in Italy, it is so widespread that it alone accounts for more than 20% of the irrigated cropland [42]. Refer to Appendix A for an overview of the used data and the pre-processing employed to run WaterCROPv1 and WaterCROPv2 for the above-mentioned analyses.

### 3.1. Validation

3.1.1. *Comparison with Previous Version, WaterCROPv1*

To analyze the impact of the updates introduced in version 2, both WaterCROPv1 and WaterCROPv2 were run with the same crop and irrigation data. The used data—provided by "Agricultural Census" [42] (Table A1 in Appendix A)—refer to maize cultivation in Italy in 2010.

The comparison was run analyzing the maize irrigation water demand ($I_{mz}$), defined as the cumulative sum of the daily volume required to keep the soil above a chosen water content, $WC_{th}$, over the growing period. In this case, $WC_{th}$ was set to critical level $WC_*$. In Figure 5a, the scatterplot compares WaterCROPv1 and WaterCROPv2 cell by cell. The two versions display general consistency as the points tend to align along the bisector. However, relevant differences due to the modeling improvements emerge clearly. In particular, it is visible how the relative importance of the newly added processes (leakage and the shutting down of the evapotranspiration process) and the use of hourly precipitation time series play a major role in the differences between the versions. In fact, the dot distribution exhibits a smaller dispersion for higher irrigation water values, a condition corresponding to reduced relevance of the added physical processes. High irrigation demands are due to low precipitation that only rarely triggers leakage and evapotranspiration shutting down. On the contrary, v1 and v2 show weaker correlations for lower water demand values, which correspond to high precipitation levels, when leakage and shutting down play a relevant role. Finally, notice the significantly lower number of zero irrigation water demands according to the improved model. This result demonstrates the importance of describing precipitation on an hourly scale rather than distributing monthly values uniformly. This allows capturing periods of water shortage even in regions characterized by high mean monthly precipitation.

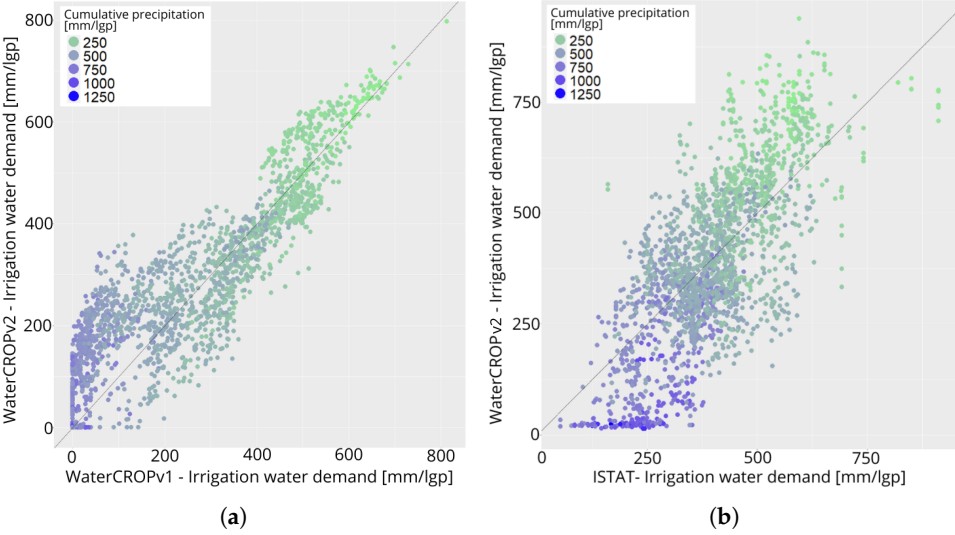

**Figure 5.** Scatterplots comparing, cell by cell, the irrigation water demand of maize in 2010 in Italy computed by (**a**) WaterCROPv2 and WaterCROPv1 and (**b**) WaterCROPv2 and ISTAT database. The black line corresponds to the bisector. The color of the dots varies gradually from green (scarce precipitation) to blue (abundant precipitation) according to the values of cumulative precipitation over the growing season in the cell.

3.1.2. *Comparison with Independent Data*

In Figure 5b we tested WaterCROPv2's consistency with irrigation values provided by the Italian National Institute of Statistics, ISTAT [42]. In this case, the $I_{mz}$ values provided by WaterCROPv2 correspond to the case of irrigation up to field capacity ($WC_{th} = WC_{fc}$) rather than to a critical point. It is indeed common practice among farmers to irrigate up to field capacity. The ISTAT values provided for each municipality correspond to the output of the MARSALa model [43], a model calibrated over 300 farms assumed to be representative of Italian agriculture. MARSALa is a more detailed and computationally demanding model with respect to WaterCROPv2 as it includes a section of irrigation method scheduling that requires additional input data. Furthermore, the model relied on farm-level input data on sowing and harvesting dates, as well as on locally employed irrigation systems, none of which are available in the "Agricultural Census" [42] database used to run WaterCROPv2. Due to the robustness of the sources of the input data used to run MARSALa, we assume the output values as the benchmark for the evaluation of WaterCROPv2's reliability. Due to the different complexity of the irrigation models and the accuracy of the input data, the WaterCROPv2 values are not expected to precisely match the ISTAT values but rather to exhibit a similar trend. Figure 5b confirms this expectation as dots align along the bisector. The observed relation between estimation errors and precipitation suggests a spatial distribution of the differences, which is confirmed in Figure 6a,b. Northern Italy, generally wetter (most of blue dots in Figure 5b, shows a mean underestimation of 23%. Southern Italy, on the contrary, which is generally drier (generally green dots in Figure 5b), experiences a mean overestimation of 14%. Central Italy, with locally variable differences, shows a more consistent mean and distribution, with a mean underestimation of 2%. The overestimation of WaterCROP2 can possibly be explained by the fact that, especially in dry areas, it might be hard for farmers to always irrigate up to field capacity and meet the model assumption. Lastly, note the dots in the bottom left of Figure 5b, where WaterCROPv2 and the ISTAT dataset show strong disagreement: they correspond to cells with high precipitation, mainly clustered in two specific pre-alpine areas in the northwest and northeast of Italy. The real ground slope in those areas might explain the disagreement.

In fact, runoff in sloped fields is generally larger than in flat areas. As WaterCROPv2 does not include a terrain model, the underestimation of the runoff in those areas, and the consequent infiltration overestimation, might result in smaller $I_{mz}$ values.

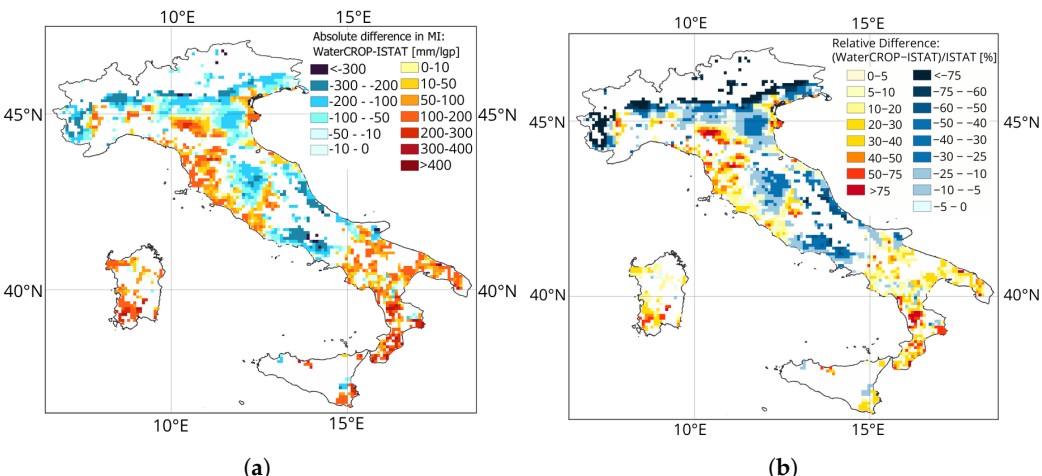

**Figure 6.** (**a**) Absolute and (**b**) relative differences in irrigation water demand of maize in 2010 computed by WaterCROPv2 and ISTAT.

### 3.1.3. *Comparison with Previous Local-Scale Studies*

The mean maize irrigation water demand estimated with WaterCROPv2 was validated against values reported in field- or local-scale studies. For instance, Katerji et al. (2013) [44] assessed the maize irrigation demand in southern Italy (Rutigliano) for the 1996–1997 growing seasons using the AquaCrop model on a private farm. Bocchiola et al. (2012) [45] investigated the irrigation requirements in the Po Plain (Persico Domico) between 2001 and 2010 using CropSyst. Casa et al. (2009) applied remote sensing data combined with the FAO method [30] to evaluate the mean irrigation needs in the Pontina Plain in the 1996–2001 period [46]. Similarly, Todisco et al. (2007) [47] analyzed the average maize water requirements in several locations in Umbria (central Italy) over the 1951–2004 period using CropSyst. In Figure 7, the comparison between the values reported in the above-mentioned studies (*I*-comp) and those obtained with WaterCROPv2 (*I*-WC) is presented. In the insets, the black-outlined rectangles mark the WaterCROPv2 pixels that overlap with the areas analyzed in the previous studies, with the corresponding $I_{mz}$ values shown directly within the pixels. The *I*-WC value shown next to each inset represents the average of the pixel-scale $I_{mz}$ values. All the comparisons considered highlight the good performance of our model despite the smaller number of parameters used and the national scale of our analysis. Only the case of the Persico Domico area (see panel Figure 7a) shows a significant difference: 120 mm/lgp estimated by WaterCROPv2 versus about 200 mm/lgp reported in [45]. However, it should be noted that the *I*-comp value refers to a single farm (shown in red in the panel), whereas the WaterCROPv2 estimate integrates contributions from several municipalities and fields. Finally, in panel Figure 7b, all the areas falling within the Pontine Plain are colored, but the crop distribution reported by [46] shows that corn is mainly located in the municipality of Latina (shown in light brown in the panel). For this reason, the numerical comparison with our model refers to this specific area.

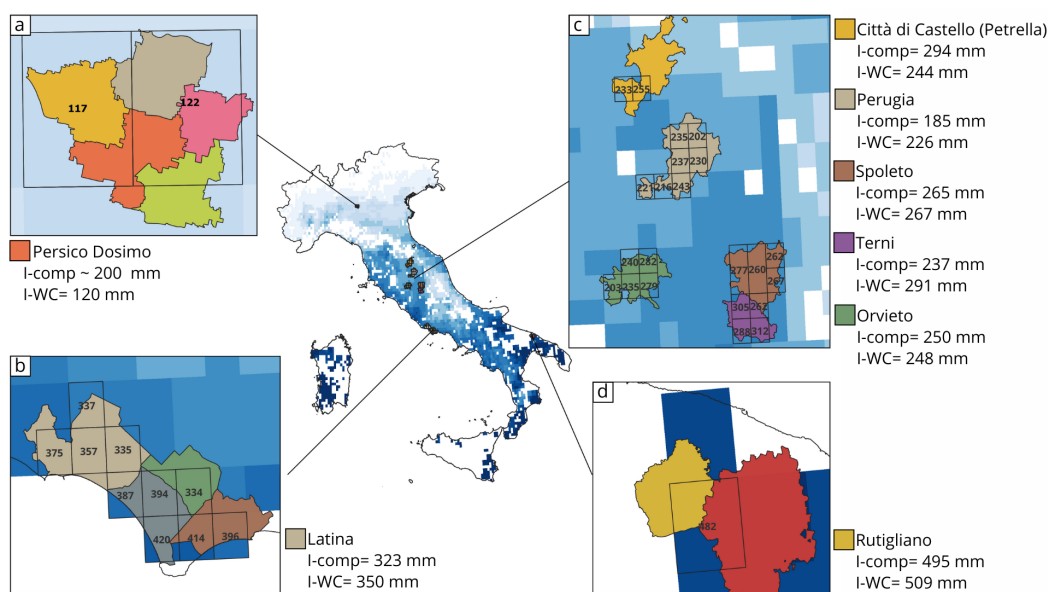

**Figure 7.** Comparisons between irrigation requirements estimated at the national scale with WaterCROPv2 (*I*-WC) and irrigation requirements evaluated in previous studies at field or local scale (*I*-comp): (**a**) Po Plain (Persico Domico) [45], (**b**) Pontina Plain [46], (**c**) Umbria region [47], (**d**) Puglia region (Rutigliano) [44].

### 3.2. Examples of Model Application

### 3.2.1. Water Demand Assessment

In this section, we present some possible applications of WaterCROPv2 as a tool to evaluate the irrigation water demand both in the current state and scenarios of interest.

We used WaterCROPv2 to assess the mean present state of maize irrigation water demand ($I_{mz}$) in Italy. Thus, we employed crop, soil, climate, and irrigation data as specified in Appendix A and assumed an irrigation threshold up to critical level ($WC_{th} = WC_*$). The used precipitation record covers the period from 2005 to 2015 as this timespan is considered representative of the Italian climate. In fact, Italy in those years experienced fairly large oscillations in cumulative precipitation, with dry (e.g., 2010–2011) and wet seasons (e.g., 2006–2007) [48]. The mean present state was evaluated by averaging the $I_{mz}$ values for each simulated year. As indicated in Equation (A3), the mean $I_{mz}$ is defined as maize blue water demand ($I_b$) divided by weighted irrigation system efficiency. For each cell, the weight of each irrigation system is based on the relative number of hectares irrigated with that system with respect to the total amount of irrigated hectares. Figure 8a,b show $I_{mz}$ in cubic meters and millimeters, respectively, presenting two different perspectives. The cubic meter perspective reveals the total actual amount of water used for maize in a specific cell. However, it does not provide information on the areal density of such amount of water. The millimeters, instead, corresponding to the normalization of the m$^3$ over the cultivated areas, provide this detail. Millimeters highlight which areas are more suitable for maize cultivation due to the local climate. The evident gradient in millimeter demand from the wetter north to the drier south points out how the local climate strongly influences the water demand. Two good examples are Pianura Padana (red rectangle) and Salentino (green rectangle). In Pianura Padana, climate and soil are suitable for maize growth, with $I_{mz}$ generally below 120 mm/lgp, leading to a large maize cultivation and high volumetric water demand (above 200,000 m$^3$/lgp). In contrast, Salentino area has a much drier climate, resulting in high millimeter demand and less extensive maize cultivation, thus leading to lower m$^3$ demand. It is noteworthy that most areas with the highest $I_{mz}$ correspond to regions where maize is the primary crop (Figure A1a, Appendix A), making maize the pivotal crop in those areas. The areas showing high or small water demand in both maps

are areas the attention of decision-makers should be driven to. In fact, the dark-blue cells in both maps are regions where the present extensive cultivation is actually located in areas poor in rainwater, and thus crop shifts should be put in place. Light-gray cells, instead, spotlight zones where maize cultivation should be developed (focusing only on the maize perspective) as it is not yet largely present despite the favorable climate.

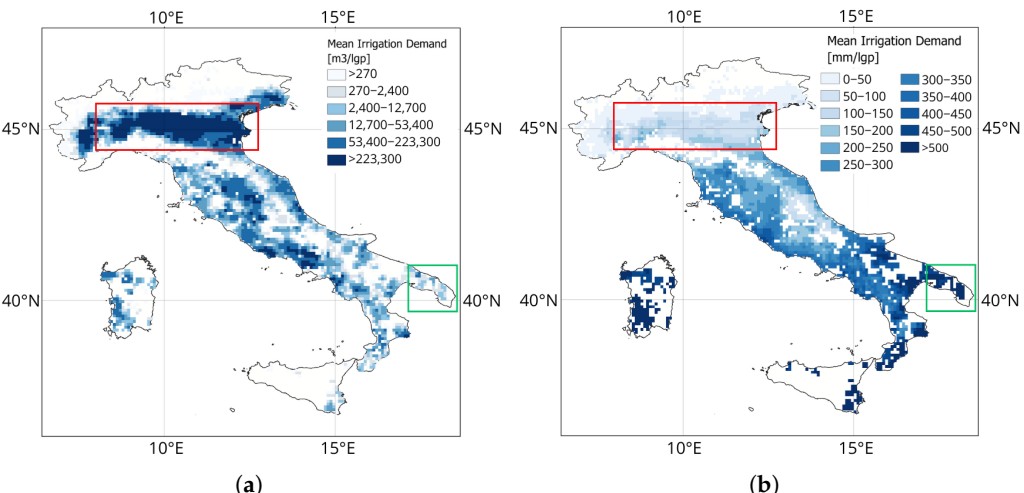

**Figure 8.** Map of modeled mean irrigation demand by maize in Italy expressed in (**a**) cubic meters; (**b**) millimeters. The red rectangle spotlights the Pianura Padana region, while the green one encloses the Salentino region.

The key sources of uncertainty, cell by cell, in the $I_{mz}$ evaluation are the irrigation system heterogeneity and the efficiency. While it is not possible to assess a range of local efficiencies in the absence of more information, a reasonable computation of the uncertainty in $I_{mz}$ due to the heterogeneity and the relative relevance of the irrigation systems can be carried out instead. Thus, two other possible values of $\alpha$ were considered: they correspond to the scenarios of upper (lower efficiency) and lower (higher efficiency) boundaries of $I_{mz}$. They can be defined based on whether maize was assumed to be irrigated primarily using less or more efficient irrigation systems. The upper $\alpha$ value was computed covering the irrigated area from the least efficient irrigation system to the most efficient one until exhaustion. For the lower $\alpha$ value, the irrigation systems were applied from the most efficient to the least efficient. Figure 9a,b display the offsets of the highest and lowest $I_{mz}$ values, respectively, as percentage deviations with respect to mean values. The correspondence of zones with the smallest deviations—[0–5%]: light yellow for the upper limit and light blue for the lower limit—among the figures indicate higher reliability of $I_{mz}$ values due to smaller heterogeneity regarding the irrigation systems in those areas. These cells account for almost 15% of the cells. In the majority of the cells (92%), the lower $I_{mz}$ deviates less than 25% from the mean, while in 67% of the cells the higher $I_{mz}$ is at most 30% larger than the mean. This skewness regarding the distribution around the mean is due to the fact that micro-irrigation, the most efficient system, covers less than 18% of the irrigated areas, while flow irrigation, one of the least efficient methods, is used on over 30% (Table 2). If local information on the applied irrigation system were available, the uncertainty would be limited to the irrigation efficiency values, whose uncertainty has a smaller impact on $I_{mz}$ computation at the national level.

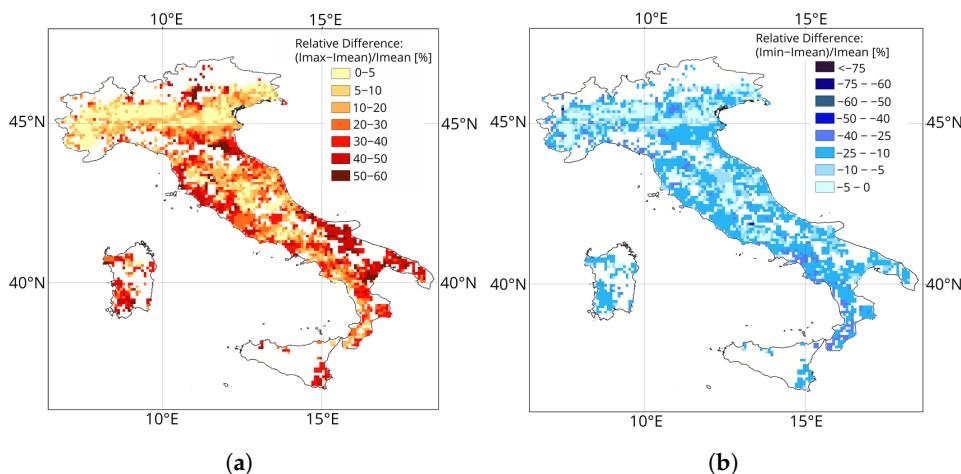

(**a**)　　　　　　　　　　　　　　　(**b**)

**Figure 9.** Relative difference in modeled (**a**) maximum and (**b**) minimum irrigation water demand with respect to the mean irrigation requirement.

### 3.2.2. *Scenario Analysis*

A key use of agro-hydrological models, such as WaterCROPv2, is to evaluate irrigation water demand. However, even though they are generally used as assessment tools, they can also be applied as modeling tools. In this sense, they are key instruments to analyze scenarios and propose alternatives to the current state. For instance, it is possible to analyze the effect of changing irrigation thresholds (e.g., irrigation up to fractions of field capacity) or timing (e.g., biweekly or weekly) to study the impact of different irrigation systems and to investigate the effects of climate change. These scenarios are fundamental to make the right decisions on the management and planning of water resources. As an example, we report here (Figure 10) the hypothetical scenario where maize is irrigated solely by micro-irrigation. It can be useful for critical analysis, localization, and quantitative evaluation of the water savings of potential investments in new micro-irrigation systems. Figure 10 shows that, at the national level, the mean water saving is 21%, and in 75% of the cells it goes up to 28% (59 mm/lgp). In some areas, it can have even larger effects: it is evident how the strongest potential water-use reduction of 30–40% is mostly located close to the Alps and the southern Apennines, where flow irrigation is now predominant due to the abundance of water (Figure A1b, Appendix A). This information becomes particularly valuable when considering future temperature and precipitation projections. Studies on climate change impacts foresee in the areas close to the mountains an increase in mean temperature of 4 °C and a shift in precipitation distribution towards winter, causing an increase in drought spells in summer [49]. Thus, the increased evapotranspirative water demand caused by the higher temperatures will find even less support in the precipitation water, increasing the stress on the irrigation water.

It is also possible to evaluate how incentives on the increase in micro-irrigation use (where it is already present) can still bring 5–10% savings. This is important regarding regions such as Salentino (recall the green rectangle in Figure 8a, very scarce in rainwater). Figure 11 shows both the preferred irrigation system (indicated by color) and the amount of water (indicated by shades) currently used in each cell where maize is cultivated. This paired information allows one to better understand where potential investments would have the greatest impact as it spotlights where large amounts of water are conveyed on the field with inefficient systems. For example, the dark-blue and dark-green areas, corresponding to central regions of Pianura Padana (recall the red rectangle in Figure 8a, representing key target areas for implementing the most effective changes in irrigation systems, resulting in the most water-demanding zones and the most inefficient irrigation systems on the

ground). At the same time, it is possible to evaluate the feasibility of changing irrigation systems: investments in micro-irrigation are more likely to occur regarding fields without existing equipment rather than in areas already equipped with sprinklers.

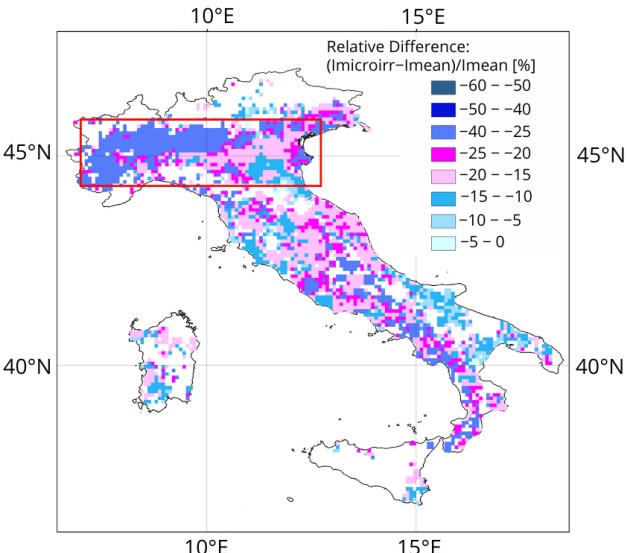

**Figure 10.** Percentage difference in modeled irrigation water demand when only micro-irrigation is used compared to the mean irrigation requirement under current mixed irrigation systems.

**Table 2.** Extension of the areas irrigated with a specific irrigation system in Italy, their efficiencies [50], and their relative relevance at national scale.

| Irrigation System | Coverage [$10^3 \cdot$ha] | % | Efficiency $\eta$ |
|---|---|---|---|
| Submersion | 221.0 | 9 | 0.25 |
| Micro-irrigation | 423.0 | 17 | 0.9 |
| Flow and Lateral infiltration | 748.4 | 31 | 0.55 |
| Sprinklers | 958.5 | 40 | 0.75 |
| Other | 68.4 | 3 | 0.7 |

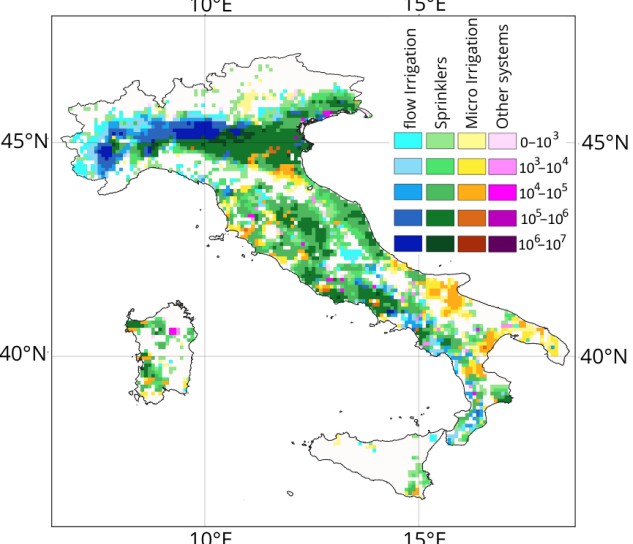

**Figure 11.** Classification of $I_{mz}$ according to irrigation system (flow irrigation in blue, sprinklers in green, and micro-irrigation in yellow) and volumetry class (reported to the right of the color legend in m$^3$).

## 4. Sensitivity Analysis

A sensitivity analysis was carried out to evaluate the impact of data uncertainty on the estimation of water demand, $ET_b$. To this aim, selected quantities were perturbed one at a time to assess their individual impact on the model. To compare the sensitivity of $ET_b$ to the different quantities, $q$, the normalized sensitivity index, $SI_q$, was applied [39]:

$$SI_q = \frac{\Delta ET_{b,q}}{ET_{b,q}} \bigg/ \frac{\Delta q}{q} \tag{12}$$

where $\Delta ET_{b,q}$ is the variation that $ET_b$ undergoes due to the quantity perturbation, $\Delta q$. We focused on the sensitivity with respect to four key input parameters: maximum rooting depth, $Zn_{max}$, relative soil water content at field capacity, $\theta_{fc}$, interception, $T$, and precipitation, $P$. Rooting depth and interception were considered to test the impact of simplifying assumptions in the model; in fact, WaterCROPv2 root growth throughout the entire season does not depend on environmental conditions that may alter the potential growth, whereas, in reality, growth rates vary depending on external factors such as soil compaction, water availability, and microbial activity, and an interception independent of canopy extension. Variations in $\theta_{fc}$ reflect the challenges of characterizing soil properties at the cell scale as they vary within fields and across soil profiles due to agricultural practices (e.g., compaction). Finally, precipitation can display highly localized patterns (particularly in spring and summer) that may not be captured by spatially distributed datasets. Positive and negative variations of 5% were applied to analyze the model's response in terms of both the magnitude and direction of change (see Table 3).

Figure 12 illustrates the variability of SI$q$ for each perturbed quantity. Lighter and darker shades correspond to negative and positive 5% variations in the quantities, respectively. The extent of the boxplots reflects the spatial variability of SI across Italy. It is evident that soil properties, $\theta fc$, and precipitation, P, have the greatest influence, while rooting depth, $Zn_{fin}$, and the interception threshold, T, exhibit smaller effects. Additionally, negative and positive perturbations produce effects that are similar in magnitude but opposite in direction: on average, soil properties and precipitation have SI values equal to $\pm 0.86$ ($-0.79/+0.93$) and $\pm 0.8$ ($+0.84/-0.75$), respectively, while $Zn_{fin}$ and $T$ $\pm 0.20$ and $\pm 0.17$, respectively. The magnitude of the variations at the local scale is consistent across scales, being comparable to the national-level evaluations (Table 3). Overall, the stability analysis indicates that the model is robust as changes in the input quantities produce output variations smaller than the perturbations themselves (i.e., $SI < 1$).

**Table 3.** Sensitivity analysis: tested quantities and their associated variations in water demand.

| Variable | Description | Initial Value | Variation | Final Value | National Water Demand Variation | National SI$_v$ |
|---|---|---|---|---|---|---|
| $Zn_{fin}$ | Maximum rooting depth | 1 m | $\pm 5\%$ | 1.05/0.95 | $-0.83\%/+0.84\%$ | $-0.166/+0.168$ |
| $\theta_{fc}$ | Relative soil water content at field capacity | 0.275 (silt–clay–loam) | $\pm 5\%$ | 0.28875/0.26125 | $+4.44\%/-4.13\%$ | $+0.88/-0.826$ |
| | | 0.225 (loam) | $\pm 5\%$ | 0.23625/0.21375 | | |
| | | 0.125 (loam–sand) | $\pm 5\%$ | 0.13125/0.118 | | |
| T | Interception | 0.5 mm | $\pm 5\%$ | 0.525/0.475 | $+1.03\%/-1.00\%$ | $+0.206/-0.2$ |
| P | Precipitation | site- and hour-specific | $\pm 5\%$ | site- and hour-specific | $-5.10\%/+5.53\%$ | $-1.02/+1.106$ |

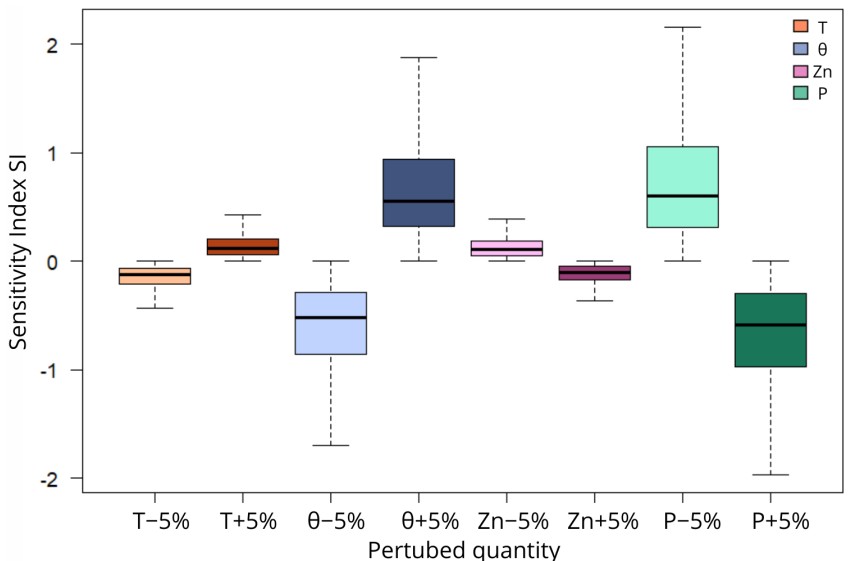

**Figure 12.** Boxplots of the sensitivity index, *SI*, of the water demand evaluated with WaterCROP model. The x-axis shows the perturbed quantities (both positive and negative): interception threshold, $T$, relative water content at field capacity, $\theta_{fc}$, maximum rooting depth, $Zn_{fin}$, and precipitation, $P$.

## 5. Conclusions and Recommendations

The aim of this work was to provide a physically based agro-hydrological model that could balance the reliability of the evaluation of irrigation water demands at regional scales with the simplicity of required inputs and short computational time demand. Water-CROPv2 possesses these characteristics. By building on the original WaterCROPv1 model, it maintains the use of readily accessible data but enhances the precision of evapotranspiration calculations through the inclusion of hourly resolution, daily plant physiology, rainwater interception, and leakage dynamics, all critical factors for regions with diverse climates and irrigation practices. However, the authors acknowledge that the model has some limitations under extreme conditions as it does not account for processes such as reduced root and canopy growth under severe stress, soil hysteresis, or water stress caused by excess water. In addition, it was specifically developed for herbaceous and seasonal crops and is not directly applicable to trees or perennials. Nevertheless, WaterCROPv2 provides improved irrigation demand estimates compared to the previous version and shows good to excellent agreement with reference data and previous studies at the local scale. This consistency demonstrates that, despite its lower complexity and non-site-specific input data, the model is able to reliably estimate crop water needs. Furthermore, sensitivity analysis with respect to key quantities demonstrated the robustness of the proposed model's outputs.

The application of WaterCROPv2 to maize cultivation in Italy was used to show the potential of the model to be applied as a valuable decision-making tool for water managers and policymakers. Specifically, the model offers an efficient approach to assess the current irrigation needs as well as to develop future sustainable water management practices. The evaluation of the mean irrigation water demand of maize (2005–2015) allowed us to identify areas where maize production is either suitable or unsustainable due to the climatic conditions and to spotlight the most water-demanding zones. The possibility to include different precipitation patterns and simulate irrigation scenarios (e.g., irrigation systems and timings) allowed us to evaluate the effects of these variables on water demand and make investment decisions accordingly. The presented example scenario showed the impact of switching maize irrigation entirely to micro-irrigation, resulting in a mean national reduction of 21% in water use with potential savings of up to 30–40% in areas

characterized by high water availability. Furthermore, by intersecting water demand maps with irrigation system maps, it was possible to gain critical insights into the areas that would benefit most from investments in upgrading irrigation systems.

In conclusion, from the perspective of increasing water scarcity and the growing need for efficiency, WaterCROPv2 provides a practical way to assess the sustainability of agriculture in a given area and identify where adjustments are most urgent and effective. The model's simplicity allows evaluations to be undertaken without demanding specialized expertise while safeguarding process transparency and mitigating the 'black-box' effect characteristic of more complex approaches.

Regarding the Italian case study, future research should extend the analysis of irrigation water demand to encompass all major crops and compare total agricultural demand with local water availability. Such an approach would help to address issues related to groundwater and surface water exploitation for agricultural purposes, as well as evaluate variations in demand driven by climate change and the expected pressure on water resources. These assessments could also contribute to the water–energy–food nexus by reducing agricultural water demand and providing evidence to support investments in crop diversification or improvements in irrigation systems.

**Author Contributions:** Conceptualization, N.C.T., M.T., F.L. and L.R.; methodology N.C.T., M.T., F.L. and L.R.; software, N.C.T. and M.T.; validation, N.C.T.; writing—original draft preparation, N.C.T.; writing—review and editing, N.C.T., M.T., F.L. and L.R.; supervision, M.T., F.L. and L.R. All authors have read and agreed to the published version of the manuscript.

**Funding:** This study received funding from the European Union Next-GenerationEU (PIANO NAZIONALE DI RIPRESA E RESILIENZA (PNRR)—MISSIONE 4 COMPONENTE 2, INVESTIMENTO 1.4—Avviso n. 3138 del 16/12/2021, Codice Programma CN00000022).

**Data Availability Statement:** The data analysis was performed using [51]. The R scripts and the data used to execute the analyses in the paper can be found at https://doi.org/10.5281/zenodo.14217709.

**Acknowledgments:** This study was carried out within the "National Research Centre for Agricultural Technologies—AGRITECH" and received funding from the European Union Next-GenerationEU. This manuscript reflects only the authors' views and opinions; neither the European Union nor the European Commission can be considered responsible for them.

**Conflicts of Interest:** The authors declare no conflicts of interest relevant to this study.

## Appendix A. Data Pre-Processing

To study the water demand of maize in Italy, accurate research was carried out among databases and literature data to build the best dataset combining availability, accuracy, handiness, and reliability (Table A1). According to the resolution of the selected datasets, the case study was run at $5 \times 5$ arc-min (pixel of $\sim 8 \times 8$ km$^2$ on average in Italy) cell spatial resolution for the year 2010. This reference year was selected due to data constraints: detailed information at the municipal scale was only provided by the 6th Agricultural Census carried out by the National Institute of Statistics, ISTAT [42], which refers to 2010.

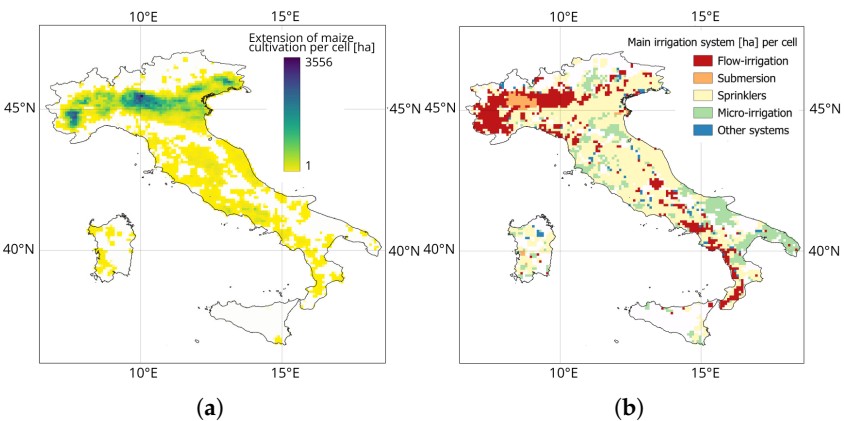

**Figure A1.** Maps of (**a**) the extension of maize cultivation in each cell; (**b**) the most extensively adopted irrigation system. Data refer to year 2010.

**Table A1.** Datasets adopted for the model application to maize in Italy. [1] [52], [2] [42], [3] [53], [4] [30], [5] [54], [6] [55], [7] [56], [8], [9] [57], [10] [50].

| Data Type | Variable | Dataset | Temporal Resolution | Spatial Resolution |
|---|---|---|---|---|
| Crop | Potential evapotranspiration | CRU [1] | month | 0.5° |
| | $p$ fraction [3] | - | year | 5 arc min |
| | $k_c, Dr, Zn_{ini}, Zn_{max}, d_s, d_h$ | FAO-56 [4] | - | - |
| Soil | Available water content | Harmonized World [5] Soil Database v1.2 | year | 5 arc min |
| | Pedologic characteristics [6] | - | - | - |
| | Soil type | LUCAS [7] | year | 500 m |
| Climate | Climate zones | PAMDataset [8] | year | 5 arc min |
| | Precipitation | ERA5 [9] | hour | 0.25° |
| Irrigation | Cultivated areas | CensimentoAgricoltura2010 [2] | year | municipality |
| | Irrigated areas | CensimentoAgricoltura2010 [2] | year | municipality |
| | Irrigation system | CensimentoAgricoltura2010 [2] | year | municipality |
| | Irrigation system efficiency [10] | - | - | - |
| | Municipalities extensions | ConfiniAmministrativi2010 [2] | year | municipality |

**Crop data.** The daily reference evapotranspiration values were defined as $1/30$ of the monthly values provided in the CRU dataset [52]. Given the lack of localized characterization of maize in terms of growth ($k_c, Zn_{ini}, Z_{max}$) and response to water stress ($Dr_*$), we referred to general values provided by [58] (Table A2).

**Table A2.** Crop parameters for maize [58].

| $Zn_{ini}$ | $Zn_{max}$ | $Dr_*$ | $kc_{ini}$ | $kc_{mid}$ | $kc_{end}$ |
|---|---|---|---|---|---|
| 0.3 | rf: 1.7 irr: 1 | 0.55 | 0.3 | 1.2 | 0.5 |

Differently, the grid dataset of sowing and harvesting dates available for cultivated areas in 2000 [59] allowed us to take into account the quite high climatic heterogeneity of the Italian territory. By employing a resampling process, we were able to define local sowing, $d_S$, and harvesting dates, $d_H$. Where cultivated areas existed in 2010 but not in 2000, and consequently lacked corresponding dates, we performed a linear spatial interpolation based on the nearest 24 neighboring cells. Through the evaluation of $d_S$ and $d_H$, we defined

*lgp*: the length of each *g*-th growing stage. The length of each stage varies according to the climate zone and was computed as a fraction, *p*, of the total growing period length [53]

$$lgp_g = p_g\dot{(}d_S - d_H) \tag{A1}$$

**Soil data.** To categorize the Italian soil into soil classes, we relied on the 500 × 500 m resolution dataset LUCAS [56]. As this resolution is higher than 5 arc-min, an upscale was run. The type of soil attributed to the 5 arc-min cell was the most recurring one among the 500 × 500 m cells falling inside the larger cell. To obtain a consistent result, the behavior of the different classes of soil to leakage was compared. The aim was to check whether some classes could be merged into one and, thus, correctly assign a soil type to each 5 arc-min cell. The classes that showed similar trends have pedological differences that can be considered negligible for the aim of this study. Figure A2 shows the different behaviors of the eight soil classes. It is evident how the eight classes group into four clusters. For the sake of simplicity, even though the sand–loam class is a cluster on its own, it was associated with loam–sand as very few cells fell into this class, and the remaining classes were traced back to three classes: silt–clay–loam, loam, and loam–sand.

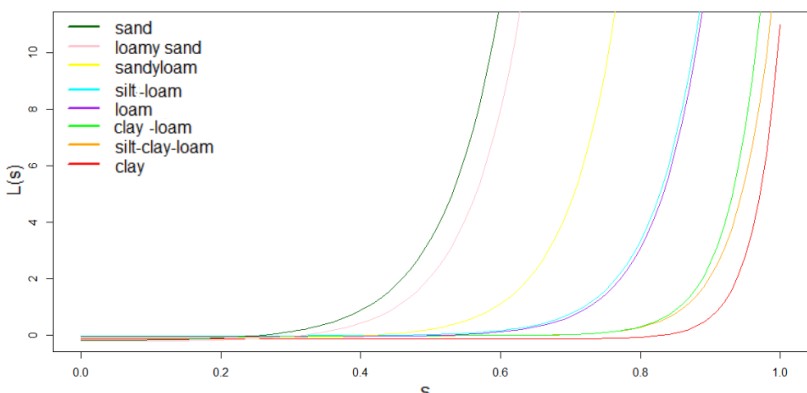

**Figure A2.** Leakage *L* according to the type of soil as a function of *s*, the relative soil moisture: sand in black, loam–sand in pink, sand–loam in yellow, silt–loam in light blue, loam in purple, clay–loam in green, silt–clay–loam in orange, and clay in red.

Typical values of hydraulic soil characteristics, for each class, are reported in Table A3.

**Table A3.** Soil characteristics [55].

| Soil Class | $K_s$ [mm/h] | b | $\theta_{sat}$ | $\theta_{fc}$ |
|---|---|---|---|---|
| Silt–Clay–Loam | 0.0612 | 7.75 | 0.477 | 0.275 |
| Loam | 0.2502 | 5.39 | 0.451 | 0.225 |
| Loam–sand | 5.628 | 4.38 | 0.401 | 0.125 |

**Climate data.** Local climate conditions were defined by the climate zones provided in the PAMDataset [54], and the hourly precipitation data were taken from the reanalysis ERA5 dataset [57]. Geographical coordinates correspond to the location of the centroid of each grid cell.

**Irrigation data.** Data on the extension of irrigated areas were available for 2010, only in listed form and by municipality thanks to the agriculture census run by the National Institute of Statistics (ISTAT) [42]. To retrieve spatial information (with 5 arc-min resolution) on maize, the listed data were first combined with the spatial location of each municipality and then rasterized in grid cells (Figure A3). The rasterization was necessary because

more than one municipality fell within the same grid cell and each municipality fell within multiple grid cells.

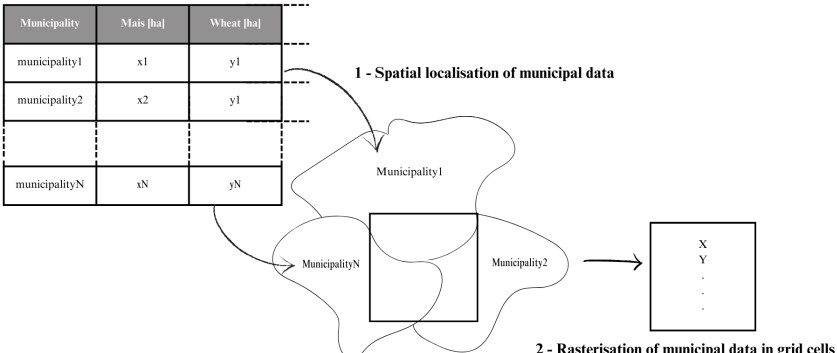

**Figure A3.** Pre-processing steps to obtain the spatial distribution of irrigated areas. The first step requires to attribute to each municipality, spatially located across Italy, its value of irrigated areas. The second step rasterizes into grid cells the information according to Equation (A2).

Lacking details at the sub-municipal scale, we assumed the maize irrigated area of each municipality $ha_m$ to be homogeneously distributed within each municipality $m$. Hence, for each grid cell, $c$, the maize irrigated area $ha_c$ [ha] was computed assuming that (i) the percentage of the municipality extension within the grid cell could be directly applied to the irrigated area, (ii) $ha_c$ to be homogeneously distributed within the grid cell, and (iii) $ha_c$ to be defined as the sum of the percentages of irrigated areas of the *m-tot* municipalities that fell within the grid cell. It follows that

$$ha_c = \sum_{m=1}^{m-tot} ha_m \frac{A_{m,c}}{A_m} \tag{A2}$$

where $ha_m$ stands for the maize irrigated area within the $m$-th municipality, $A_m$ for the municipality area, and $A_{m,c}$ for the municipality extent within the grid cell.

Unfortunately, the CensimentoAgricoltura2010 database provides, for each crop and municipality, data on the amount of irrigated hectares but not the specific irrigation system used for each crop. This did not allow us to define a local maize-specific value for the inefficiency factor $\alpha$.

To overcome this gap, information about the typical irrigation systems used for maize in each cell was intersected with information on the available irrigation systems in each municipality. It was thus possible to define, for each cell, a plausible mix of used irrigation systems. Typically, in Italy, maize is irrigated with flow and lateral infiltration, sprinklers, and micro-irrigation systems, and for their efficiencies we referred to data published by the Italian Ministry of Agriculture Food Sovereignty and Forestry [50] reported in Table 2.

Hence, for each cell, $c$, the inefficiency factor $\alpha$ was computed as

$$\alpha_c = \sum_y \left( \frac{1}{\eta_y} \frac{ha_y}{ha_{tot}} \right) \tag{A3}$$

where $y$ stands for each irrigation system used in the cell, $\eta_y$ for the corresponding efficiency, $ha_y$ for the hectares irrigated with the irrigation system $y$, and $ha_{tot}$ for the total amount of irrigated hectares within the cell.

## Appendix B. Trends of Root Growth and Crop and Stress Coefficient

Figure A4 reports the trends of $Zn$ and $k_c$ during the growing stages of the growing season. Figure A5, instead, shows the trend of $k_s$ as function of the soil water content $WC$.

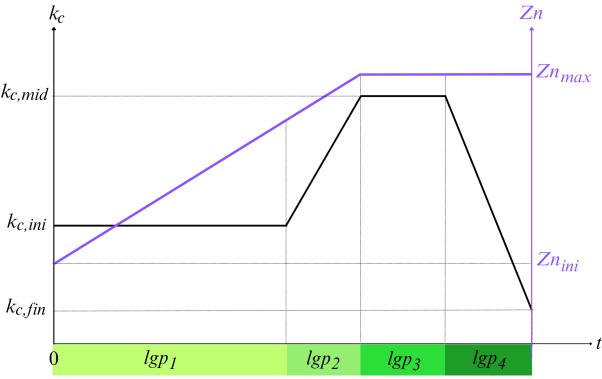

**Figure A4.** Qualitative description of the crop coefficient $k_c$ (in black) and the elongation of the roots $Zn$ (in purple) during the growing stages $lgp$s of the growing season.

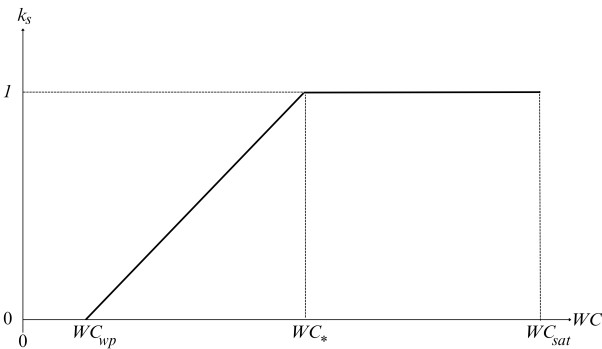

**Figure A5.** Qualitative description of the crop coefficient, $k_s$, according to the soil water content.

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
