# Peer review of "Evaluating Country-Scale Irrigation Demand Through Parsimonious Agro-Hydrological Modeling"

_hydrology, doi:10.3390/hydrology12090240_

Round 1

Reviewer 1 Report

Comments and Suggestions for Authors
  1. At present, numerous agro-hydrological models have been developed. Many of these models have a wide range of applications. It is necessary to compile a list of mainstream models for classified comparison. It will help highlight the applicability of WaterCROPv2 developed by the authors.
  2. The authors mentioned that their Model Version 2.0 in this paper was developed from Version 1.0 in Reference [36] in 2015. However, I could not find this model in Reference [36]. It is recommended that the improvement items on WaterCROPv1 be elaborated in a clear, point-by-point manner.
  3. Regarding data: The model validation utilized data of 2010, while model application employed data from 2005 to 2015. Given that the current year is 2025, why were data from such earlier years selected? Would deviations occur in terms of accuracy or other performance metrics if post-2015 data were used for model application?
  4. Could other commonly used models be employed to analyze the same dataset? By comparing the data analysis results of different models, a comprehensive assessment of the model proposed by the authors could be conducted.

Author Response

We would like to thank the Reviewer for the constructive review of the manuscript. Below, we provide detailed responses  to each of Reviewer’s comments.

[comment 1]:At present, numerous agro-hydrological models have been developed. Many of these models have a wide range of applications. It is necessary to compile a list of mainstream models for classified comparison. It will help highlight the applicability of WaterCROPv2 developed by the authors.

[response1]: We thank the Reviewer for the suggestion. In the revised version of the manuscript, we have included in the Introduction a list of six additional agro-hydrological models of varying complexity and application (see line 62), and we have also added a comparison with the results obtained from other models (see ‘3.1.3. Comparison with previous local-scale studies 296’ subsection in the ‘3. Results’ section of the revised manuscript).

[comment 2]:The authors mentioned that their Model Version 2.0 in this paper was developed from Version 1.0 in Reference [36] in 2015. However, I could not find this model in Reference [36]. It is recommended that the improvement items on WaterCROPv1 be elaborated in a clear, point-by-point manner.

[response2]: We thank the Reviewer for highlighting this point. The earlier version of the model (v1), introduced in 2015 and described in the cited publication, was not explicitly labeled as “WaterCROPv1” at that time. Nevertheless, it is the parent model from which WaterCROPv2 was developed. To address the Reviewer’s concern, we have clarified this connection in the revised manuscript (lines 84–91) and have now included a point-by-point description of the improvements made in WaterCROPv2 compared with the earlier version.

[comment 3]:Regarding data: The model validation utilized data of 2010, while model application employed data from 2005 to 2015. Given that the current year is 2025, why were data from such earlier years selected? Would deviations occur in terms of accuracy or other performance metrics if post-2015 data were used for model application?

[response3]:We thank the Reviewer for raising this important concern regarding the use of up-to-date data. At present, national-scale data with high spatial resolution are only available through the Italian National Institute of Statistics (ISTAT) database and no more recent datasets are publicly accessible.

We acknowledge that some deviations could occur if post-2015 data were available, particularly due to recent variations in rainfall patterns. However, we expect that results concerning Blue Water demand would not differ significantly, as ISTAT data primarily report cultivated area, which for maize has remained relatively stable over time. We also recognize that irrigation demand may have been influenced by the wider adoption of more efficient irrigation systems in recent years. This potential effect has been considered and is reflected in Fig. 10 of the manuscript.

[comment 4]:Could other commonly used models be employed to analyse the same dataset? By comparing the data analysis results of different models, a comprehensive assessment of the model proposed by the authors could be conducted.

[response4]: We thank the Reviewer for the question and the suggestion. In principle, other agro-hydrological models could be applied; however, widely used agro-hydrological models are generally more complex and are designed for local, field- or farm-scale applications. Their implementation would require detailed input data that are not available at the national scale. Conversely, global hydrological models are less detailed and therefore too coarse to adequately address regional water management objectives.

For these reasons, we chose to compare the outputs of WaterCROPv2 with irrigation water demand data reported by ISTAT, which—being derived from the detailed MARSALa model—represents the only nationwide reference available for Italy. In addition, in the revised manuscript we now include local-scale comparisons with results from studies that employed more complex models (e.g., AquaCrop [1] and CropSys [2]). These comparisons showed good to excellent agreement, further supporting the reliability of WaterCROPv2.

References:

[1] Salman, M., G.V.M.F.E.R.D.; Steduto, P. The AquaCrop model – Enhancing crop water productivity. Ten years of development, dissemination and implementation 2009–2019. FAO Water Report 2021. https://doi.org/https://doi.org/10.4060/cb7392en. 783

[2] Allen, R.; Pereira, L.; Raes, D.; Smith, M. Crop evapotranspiration - Guidelines for computing crop water

requirements - FAO Irrigation and drainage paper 56. FAO - Food and Agiculture Organization of the United

Nations 1998.

Reviewer 2 Report

Comments and Suggestions for Authors

This manuscript presents WaterCROPv2, an enhanced agro-hydrological model for estimating national-scale irrigation water demand by reconciling physical detail with practical data requirements. Incorporating hourly time resolution, canopy interception, soil-dependent leakage, diurnal evapotranspiration dynamics, and irrigation system inefficiencies, the model simulates both theoretical crop water demand and actual water volumes needed in the field. The authors demonstrate its performance through a case study of maize cultivation in Italy, including validation against the previous version and independent ISTAT data, and explore scenario analyses—most notably the potential water savings from widespread adoption of micro-irrigation—thereby illustrating the model’s utility as a decision-support tool for strategic water management.

The manuscript’s strengths include a well-balanced modeling framework that combines physical detail (hourly dynamics, canopy interception, leakage, and irrigation inefficiencies) with practically available data, and a clear country-scale application with validation and scenario exploration.

However, the following major revisions are important:

Abstract:
The abstract does not report any of the study’s results. It must clearly and concisely state the objective, scope, and key findings of the work—including quantitative outcomes where possible—so that readers understand what was done and what was learned without reading the full paper.

Introduction:
The current Introduction is overly long and fragmented; it would be sufficient if condensed to around 5–6 well-structured paragraphs. The background is not developed in enough depth, and recent relevant work on water management is missing. Key recent studies should be cited and discussed (e.g., DOI: 10.1080/03067319.2025.2482187, 10.3390/w17142136, 10.1016/j.jhydrol.2025.133948).
There remain serious concerns about the originality of the contribution. What is the fundamental difference that distinguishes this study from prior work? The manuscript should explicitly articulate how the proposed model builds upon, differs from, or advances previous studies.

Model and Data:
This section does a good job of describing the proposed model and the input datasets. However, it would be very valuable to include a flowchart or workflow diagram of the model structure and data processing pipeline to improve clarity and reproducibility.

Results:
The Results section is currently too brief and weak. All obtained results should be organized under clear subheadings. For each result, the authors need to describe the context in which it was obtained, include discussion/interpretation, and highlight any unexpected behaviors or sensitivities encountered.
Moreover, validation is insufficiently addressed. If independent validation data are available, why are the relationships between model outputs and validation data not quantitatively analyzed and reported? If no validation data exist, the manuscript must justify confidence in the model’s outputs—e.g., via statistical tests, sensitivity analyses, or uncertainty quantification. The model’s performance and robustness need to be assessed rigorously.

Discussion and Conclusions:
This section lacks substantive discussion. For example, how do the results compare and contrast with findings from previous studies? The authors must critically evaluate their own results, not merely restate them.
The manuscript should explicitly list its strengths and weaknesses, clearly state limitations, and provide concrete recommendations for future work.
A useful guiding principle: the data, methods, and results should be presented with such clarity and transparency that an independent researcher could reproduce the key findings using the same inputs and procedures.

Summary of Required Major Revisions:

  • Add results to the Abstract.
  • Expand and tighten the literature review in the Introduction; clearly emphasize the manuscript’s originality and its relation to prior work.
  • If feasible, include a workflow/model diagram in the Model and Data section.
  • Substantially rewrite the Results section: organize with subheadings, include quantitative validation, discuss findings, and assess robustness.
  • Rewrite the Discussion and Conclusions: provide critical interpretation, compare with prior literature, state strengths/limitations, and offer forward-looking suggestions.

Author Response

We would like to thank the Reviewer for the constructive review of the manuscript. Below, we provide detailed responses  to each of Reviewer’s comments.

[comment1]: The abstract does not report any of the study’s results. It must clearly and concisely state the objective, scope, and key findings of the work—including quantitative outcomes where possible—so that readers understand what was done and what was learned without reading the full paper.

[response1]: We thank the Reviewer for the comment and suggestion. In the revised version of the manuscript, we included quantitative values and findings: “[..]The model identifies optimal cultivation areas, such as the Pianura Padana, where irrigation requirements do not exceed 200 mm for the entire maize growing period, and unsuitable regions, such as Salentino, where over 500 mm of water per season are required due to local climatic conditions. In addition, estimates of the water volumes required for the current extent of maize cultivation show that the Pianura Padana region demands nearly three times the amount of water used in the Salentino area.

The model is also used to identify regions where adopting efficient irrigation technologies could lead to substantial water savings.

With micro irrigation currently covering less than 18% of irrigated land, simulations suggest that a complete transition to this system could reduce national water demand by 21%. Savings could reach 30–40% in traditionally water-rich regions that rely on inefficient irrigation practices but are expected to be increasingly exposed to temperature increase and precipitation shifts. The analysis shows that regions currently lacking adequate irrigation infrastructure stand to gain the most from targeted irrigation system investments but also highlights how incentives where microirrigation is already widespread can bring a further 5-10% saving.” See lines 20-33.

[comment2]:
The current Introduction is overly long and fragmented; it would be sufficient if condensed to around 5–6 well-structured paragraphs. The background is not developed in enough depth, and recent relevant work on water management is missing. Key recent studies should be cited and discussed (e.g., DOI: 10.1080/03067319.2025.2482187, 10.3390/w17142136, 10.1016/j.jhydrol.2025.133948).

[response2]:We thank the Reviewer for the comment and suggested references. While the cited works mainly address water management in broader, non-agricultural contexts, our study focuses specifically on irrigation and agricultural water demand. To maintain this focus, we prioritized literature directly relevant to agro-hydrological modeling. Nonetheless, we have revised the Introduction to better highlight the connection with general water management frameworks

[comment3]:There remain serious concerns about the originality of the contribution. What is the fundamental difference that distinguishes this study from prior work? The manuscript should explicitly articulate how the proposed model builds upon, differs from, or advances previous studies.

[response3]:We thank the Reviewer for the question. Our aim is to propose a model that contributes to fill the gap between the two main families of agro-hydrological models currently available: (i) models focusing on the local, field scale, and (ii) global hydrological models. The first category is highly detailed, accounting for many biotic and abiotic processes, and therefore requires extensive data to be applied. The second category, in contrast, operates at very large scales and considers only a limited set of basic physical processes. Our model is intended to lie in between: it physically represents all the key processes driving irrigation demand, relies on data that are generally accessible, and is designed to capture municipal-scale dynamics while still enabling regional and national analyses. In addition, the model is sufficiently simple to be applied also by non-experts. In the revised version of the manuscript, we have: (i) clarified more explicitly the objectives and rationale of our work; (ii) expanded the comparison with other studies; (iii) highlighted the differences and improvements with respect to our previous model, WaterCROPv1; and (iv) specified the limitations of the proposed approach in the conclusions and final recommendations.

[comment4]: This section (Model and Data) does a good job of describing the proposed model and the input datasets. However, it would be very valuable to include a flowchart or workflow diagram of the model structure and data processing pipeline to improve clarity and reproducibility.

[response4]:We thank the Reviewer for the comment and suggestion. In the revised manuscript, a flowchart highlighting the main steps included in the model has been added: see fig. 4.

[comment5]:The Results section is currently too brief and weak. All obtained results should be organized under clear subheadings. For each result, the authors need to describe the context in which it was obtained, include discussion/interpretation, and highlight any unexpected behaviors or sensitivities encountered.
Moreover, validation is insufficiently addressed. If independent validation data are available, why are the relationships between model outputs and validation data not quantitatively analyzed and reported? If no validation data exist, the manuscript must justify confidence in the model’s outputs—e.g., via statistical tests, sensitivity analyses, or uncertainty quantification. The model’s performance and robustness need to be assessed rigorously.

[response5]:We thank the Reviewer for the suggestion. In the revised manuscript, the Results section has been subdivided as follows:

3.1 Validation

3.1.1. Comparison with previous version WaterCROPv1

3.1.2. Comparison with independent data

3.1.3. Comparison with previous local-scale studies

3.2 Examples of model application

3.2.1. Water demand assessment

3.2.3. Scenarios analysis

In section 3.1.3, we present additional validations of the model results, showing the accordance with some previous studies that address the problem using different (more detailed) agro-hydrological models. The general agreement with independent data (3.1.2.) and prior local-scale studies (3.1.3) support the reliability of the model.

Furthermore, in the newly introduced Section 4 “Sensitivity analysis”, we show the robustness of the model in evaluating water demand. We present the sensitivity of the model to the variation of 4 parameters: maximum rooting depth, relative water content at field capacity, interception and precipitation.

[comment6]: This section (Discussion and Conclusions) lacks substantive discussion. For example, how do the results compare and contrast with findings from previous studies? The authors must critically evaluate their own results, not merely restate them.
The manuscript should explicitly list its strengths and weaknesses, clearly state limitations, and provide concrete recommendations for future work.

[response6]: We thank the Reviewer for the comment. In the revised manuscript, the new section “3.1.3. Comparison with previous local-scale studies” has been dedicated to the comparison of the results obtained with WaterCROPv2 and the results from previous studies run at local scale and with more complex models (e.g. AquaCrop[1] and CropSyst[2]). These comparisons revealed good agreement, underscoring the strength of WaterCROPv2 in reliably estimating crop water requirements despite its lower complexity. Furthermore, we have expanded the Conclusion and Recommendation section (lines 407–420) to explicitly discuss the model’s strengths and limitations, providing a more balanced and critical evaluation. Finally, we have included clear recommendations for future research directions (lines 441–445), as suggested.

[comment7]: A useful guiding principle: the data, methods, and results should be presented with such clarity and transparency that an independent researcher could reproduce the key findings using the same inputs and procedures.

[response7]:We thank the Reviewer for the comment and fully agree on the importance of reproducibility. For this reason, we have provided the input parameters and procedures in Appendix A and made all input data available in an open-access online repository, as stated in the “Data Availability” section. To allow better readability, we have not included these details directly in the main text.

References:

[1] Salman, M., G.V.M.F.E.R.D.; Steduto, P. The AquaCrop model – Enhancing crop water productivity. Ten years of development, dissemination and implementation 2009–2019. FAO Water Report 2021. https://doi.org/https://doi.org/10.4060/cb7392en. 783

[2] Allen, R.; Pereira, L.; Raes, D.; Smith, M. Crop evapotranspiration - Guidelines for computing crop water  requirements - FAO Irrigation and drainage paper 56. FAO - Food and Agiculture Organization of the United Nations 1998.

Reviewer 3 Report

Comments and Suggestions for Authors

This article presents the developed WaterCROPv2 model, named the agro-hydrological model, which was designed to estimate national-scale irrigation water demand while effectively balancing accuracy with practical data requirements. The article fits well with the journal theme. However, the article can not be accepted in its current form. I have the following comments to improve its quality and presentation.

1- You are going to publish in an international journal. Therefore, please start the abstract with "1-2 general statements to show the global importance of the general topic" followed by the problem statement.

2- Prepare a list of the symbols used in the article and include it at the beginning of the article or at the end of the requested "Recommendations" section.

3- Remove the last 4 lines at the end of the "Introduction" section.

4- Please add this heading and amedement:

2 Materials and Methods
Add a detailed flowchart for the methodology and write as follows:
Figure 1 shows the flowchart of the methodology used in this article.

2.1 Model
Remove all the details about the model from the introduction and include it at the beginning of this subsection or in the proper place in this subsection.

2.2 Data

5- At the end of the "Model" subsection, make a comparison with the technical elements of the two models, WaterCropv2 and WaterCropv1, to indicate the differences in WaterCropv2 that make it more accurate compared to WaterCropv1.

This could also be written close to Figure 4.a.

6- It is assumed that the y-axis of Fig. 4b is WaterCropv2 - ....."  Please revise it. I understand it is written correctly in the title, but should be corrected to on the y-axis as it is already written correctly on the y-axis of Fig. 4a.

7- Please present the results of the residuals analysis of the results/data presented in Fig. 4b to ensure the model WaterCropv2 is not biased.

8- Please write the "Conclusion" in a separate section.

9- Please add a "Recommendation" section.

10- Cite the findings of some (or at least one) related source published in 2025 and list it.

11- If possible, could you present the analysis of the results of (WaterCropv1 versus ISTAT)? (optional).

Author Response

We would like to thank the Reviewer for the constructive review of the manuscript. Below, we provide detailed responses to each of Reviewer’s comments.

[comment1]- You are going to publish in an international journal. Therefore, please start the abstract with "1-2 general statements to show the global importance of the general topic" followed by the problem statement.

[reponse1]: We thank the Reviewer for the suggestion. In the revised manuscript, we included in the abstract a general statement to better frame the context: (see lines 1-7) “Climate change is expected to reduce water availability during cropping season, while growing populations and rising living standards will increase global water demand. This creates an urgent need for national water management tools to optimize water allocation. In particular, agriculture requires targeted approaches to improve efficiency . Alongside field measurements and remote sensing, agro-hydrological models have emerged as a particularly valuable resource for assessing and managing agricultural water demand[..]”

[comment2]- Prepare a list of the symbols used in the article and include it at the beginning of the article or at the end of the requested "Recommendations" section.

[response2]-We agree with Reviewer. In the revised manuscript, we included the list of symbols.

[comment3]- Remove the last 4 lines at the end of the "Introduction" section.

[response3]-We thank the Reviewer for the suggestion. In the revised manuscript, we removed the mentioned lines

[comment4]- Please add this heading and amendment:

2 Materials and Methods
Add a detailed flowchart for the methodology and write as follows:
Figure 1 shows the flowchart of the methodology used in this article.

2.1 Model
Remove all the details about the model from the introduction and include it at the beginning of this subsection or in the proper place in this subsection.

2.2 Data

[reponse4]:We thank the Reviewer for these suggestions. In the revised manuscript, a flowchart highlighting the main steps included in the model has been added (see the new fig. 4) as well as the suggested variations on the headings. The details about the model were moved from the Introduction and placed in the “Model” subsection under the heading “Modifications of WaterCROPv2 with respect to version 1” (see lines 199-208)

[comment5]- At the end of the "Model" subsection, make a comparison with the technical elements of the two models, WaterCropv2 and WaterCropv1, to indicate the differences in WaterCropv2 that make it more accurate compared to WaterCropv1.This could also be written close to Figure 4.a.

[response5]: We thank the Reviewer for the suggestion. In the revised manuscript, the details about the differences between the two models are now reported in “Modifications of WaterCROPv2 with respect to version 1” (see lines 199-208)

[comment6]- It is assumed that the y-axis of Fig. 4b is WaterCropv2 - ....."  Please revise it. I understand it is written correctly in the title, but should be corrected to on the y-axis as it is already written correctly on the y-axis of Fig. 4a.

[response6]- We thank the Reviewer for pointing this imprecision. In the revised manuscript, fig5b (note that one figure was added) has y-axis labeled as “WaterCROPv2”

 [comment7]- Please present the results of the residuals analysis of the results/data presented in Fig. 4b to ensure the model WaterCropv2 is not biased.

[reponse7]-We thank the Reviewer for this valuable suggestion. In the revised manuscript, we have report in this answer QQ plot of the residuals (WaterCROPv2 vs. ISTAT) against a normal distribution (see attached figure). The plot shows a generally good alignment, with the exception of some cases where WaterCROPv2 underestimates water demand. This discrepancy has been addressed and discussed in Section 3.1.2. Comparison with independent data (lines 289–295). 

[comment8]- Please write the "Conclusion" in a separate section.

[reponse8]- We thank the Reviewer for the suggestion. In the revised manuscript, Conclusions comes as a separate section from the Results and it is titled “Conclusion and Recommendations” (see section 5)

[comment9]- Please add a "Recommendation" section.

[response9]-We thank the Reviewer for the suggestion. In the revised manuscript, we added the content of a “Recommendation” section to the “Conclusions”. This results in the section “Conclusion and Recommendations” (see section 5)

[comment10]- Cite the findings of some (or at least one) related source published in 2025 and list it.

[response10]-We thank the Reviewer for the suggestion. We agree and recognise the role of more recent studies; however, to the best of our knowledge there are no studies published in 2025 on the evaluation of maize water demand in Italy. However, in the revised manuscript, we added the section 3.1.3.  “Comparison with previous studies”; here, we report findings from previous studies and compare them with the results obtained with WaterCROPv2

[comment11]- If possible, could you present the analysis of the results of (WaterCropv1 versus ISTAT)? (optional).

[reponse11]-We thank the Reviewer for this suggestion. However, we believe that our work already contains a substantial number of figures (and we have added further ones in the revised manuscript). For this reason, we prefer not to include an additional figure at this stage.  

Reviewer 4 Report

Comments and Suggestions for Authors

I would like to inform you that the manuscript entitled "Evaluating country-scale irrigation demand through parsimonious agro-hydrological modeling" can be considered for acceptance. However, a minor adjustment needs to be made.

Best regards

Author Response

We would like to thank the Reviewer for their positive assessment of our manuscript and for considering it suitable for publication.

Comment: Table E1 requires significant adjustments in formatting and presentation as it is currently inadequate for inclusion in the manuscript

response: We thank the Reviewer for highlighting this issue. In the LaTeX file originally submitted, the table was formatted vertically, as its size does not allow for a horizontal layout. The discrepancy you observed is most likely due to a missing package on the editor’s side

Reviewer 5 Report

Comments and Suggestions for Authors

Please find the attached file for the comments.

Comments on the Quality of English Language

The article English can be improved. 

Author Response

We thank the Reviewer for the careful reading of our manuscript and for the valuable suggestions. In the revised version, we have incorporated all of these suggestions, modifying the text accordingly. The only point to which we prefer to respond separately is the following:

comment: why you have used only WaterCROPv2 model? Why this is the best model compared to other models or machine learning models. Add the reason and literature in the support of this model

renspose: Our aim is to propose a model that contributes to fill the gap between the two main families of agro-hydrological models currently available: (i) models focusing on the local, field scale, and (ii) global hydrological models. The first category is highly detailed, accounting for many biotic and abiotic processes, and therefore requires extensive data to be applied. The second category, in contrast, operates at very large scales and considers only a limited set of basic physical processes. Our model is intended to lie in between: it physically represents all the key processes driving irrigation demand, relies on data that are generally accessible, and is designed to capture municipal-scale dynamics while still enabling regional and national analyses. In addition, the model is sufficiently simple to be applied also by non-experts. In the revised version of the manuscript, we have: (i) clarified more explicitly the objectives and rationale of our work; (ii) expanded the comparison with other studies (please, see section 3.1.3); (iii) highlighted the differences and improvements with respect to our previous model, WaterCROPv1; and (iv) specified the limitations of the proposed approach in the conclusions and final recommendations. We recognize that machine learning approaches have recently been explored for agricultural water demand estimation (e.g., random forest[1][2], and deep learning models [3]). While these techniques can yield accurate short-term predictions when large, high-quality datasets are available, their application at national scale is still limited by issues of data availability, transferability, and interpretability. In contrast, WaterCROPv2 is fully process-based, transparent in its assumptions, and capable of operating with the data currently available at the national scale.

References:

[1]Kulaczkowski, A.; Lee, J. Harnessing the Power of Random Forest for Precise Short-Term Water Demand Forecasting in Italian Water Districts. Eng. Proc. 2024, 69, 81. https://doi.org/10.3390/engproc2024069081

[2]Z. Liu, G. Palacios-Navarro, R. Lacuesta, Agricultural big data for predicting crop water demand, Smart Agricultural Technology, Vol. 12, 2025, 101155, ISSN 2772-3755, https://doi.org/10.1016/j.atech.2025.101155.

[3]Qin, Y.-M.; Tu, Y.-H.; Li, T.; Ni, Y.; Wang, R.-F.; Wang, H. Deep Learning for Sustainable Agriculture: A Systematic Review on Applications in Lettuce Cultivation. Sustainability 2025, 17, 3190. https://doi.org/10.3390/su17073190

Round 2

Reviewer 1 Report

Comments and Suggestions for Authors

Accept in present form

Reviewer 2 Report

Comments and Suggestions for Authors

Dear Editor,

Having reviewed the authors’ detailed responses and the revised manuscript, I confirm that the work has been substantially improved and that my substantive concerns have been addressed. The abstract now reports quantitative outcomes; the introduction is streamlined with up-to-date literature; the study’s novelty is clearly articulated (bridging field-scale and global models); a workflow diagram has been added; results are reorganized with rigorous validation against independent data and prior local-scale studies; a new sensitivity analysis demonstrates robustness; the discussion explicitly states strengths, limitations, and future directions; and data/parameters are provided to enhance reproducibility. In my view, the manuscript now meets the journal’s publication standards and I recommend acceptance, subject to routine editorial checks.

Best regards,

Reviewer

Reviewer 5 Report

Comments and Suggestions for Authors

Major revisions

Round 3

Reviewer 5 Report

Comments and Suggestions for Authors

Accepted.

Comments on the Quality of English Language

The English can be improved.